# Unraveling the electrocatalytic reduction mechanism of enols on copper in aqueous media

**Zhihao Cui[1], Xing'an Dong[2], Sung Gu Cho[1], Modeste N. Tegomoh[1], Weidong Dai[2], Fan Dong[2] & Anne C. Co [1] ✉**

Deoxygenation of aldehydes and their tautomers to alkenes and alkanes has implications in refining biomass-derived fuels for use as transportation fuel. Electrochemical deoxygenation in ambient, aqueous solution is also a potential green synthesis strategy for terminal olefins. In this manuscript, direct electrochemical conversion of vinyl alcohol and acetaldehyde on polycrystalline Cu to ethanol, ethylene and ethane; and propenol and propionaldehyde to propanol, propene and propane is reported. Sensitive detection was achieved using a rotating disk electrode coupled with gas chromatography-mass spectrometry. In-situ attenuated total reflection surface-enhanced infrared absorption spectroscopy, and in-situ Raman spectroscopy confirmed the adsorption of the vinyl alcohol. Calculations using canonical and grand-canonical density functional theory and experimental findings suggest that the rate-determining step for ethylene and ethane formation is an electron transfer step to the adsorbed vinyl alcohol. Finally, we extend our conclusions to the enol reaction from higher-order soluble aldehyde and ketone. The products observed from the reduction reaction also sheds insights into plausible reaction pathways of $CO_2$ to $C_2$ and $C_3$ products.

Bio-oil derived from the pyrolysis of lignocellulose in biomass is a promising renewable feedstock for the production of commodity chemicals and fuels[1]. One of the main challenges in adopting bio-oil as a transportation fuel is their high oxygen content from hydrocarbon mixtures containing oxygenates such as carboxylic acids, ketones, and aldehydes, which necessitates additional deoxygenation strategies[2]. Based on previously reported work on the electrochemical hydrogenation of carbonyl compounds over metal electrodes[3,4], heterogeneous electrochemical deoxygenation may serve as a promising, inexpensive, modular, approach to deoxygenate carbonyl compounds in bio-oil under ambient temperature and pressure, providing a potentially economically viable pathway to refining bio-oil. The electrochemical deoxygenation pathway of aldehydes and ketones also requires understanding the reactivity of their enol tautomers, since enols are usually present at equilibrium in aqueous solution and could

serve as potential intermediates in several reactions involving simple aldehydes and ketones. Consequently, the electrochemistry of simple enols is of great relevance to the deoxygenation pathways to useful bio-oil[5]. The electrochemistry of enols, however, is underexplored because they are generally thermodynamically unfavorable in an aqueous solution, as a result, enol concentrations in solutions are usually very low. To this effect, the electrocatalytic products generated from enol conversions are hardly detected using conventional analytical methods[6].

In this work, we use acetaldehyde as a model system to elucidate the role of enols in the electroreduction of aldehydes and ketones. Acetaldehyde is one of the simplest aldehydes that has a moderate keto-enol equilibrium constant ($pK_e$) of 6.23[5] in aqueous solution. Acetaldehyde will be used as a benchmark for studying and comparing higher-order aldehydes and ketones, which have different $pK_e$ values.

[1]Department of Chemistry and Biochemistry, The Ohio State University, Columbus, OH 43210, USA. [2]Research Center for Environmental & Energy Catalysis, Institute of Fundamental and Frontier Sciences, University of Electronic Science and Technology of China, Chengdu 611731, China. ✉e-mail: co.5@osu.edu

Here, we choose polycrystalline copper (Cu) as a model electrocatalyst. First, the electrochemistry of acetaldehyde is well-established on Cu[7–11] which provides a baseline for elucidating the unique contribution of enols during the reduction reaction. Second, Cu is the only pure metal that can reduce $CO_2$ to desired $C_2$ products with substantial Faradaic efficiencies[12]. Therefore, this work will also provide valuable insight into understanding the role of enols in the selective reduction of $CO_2$ toward $C_2$ and $C_3$ products on Cu.

Here, we report an experimental methodology that allows for chemical identification with very high sensitivity, which is especially useful for identifying the products from the reduction of enols, for example, vinyl alcohol, a tautomer of acetaldehyde, that are extremely dilute in aqueous solution. First, a rotating ring-disk electrode (RRDE) setup equipped with a polycrystalline Cu disk and a polycrystalline Pt ring was employed at a high rotation rate (1600 RPM) for rapid detection of products during the reduction of acetaldehyde on a Cu disk. Next, the rotating Cu disk was coupled to an online gas chromatography-mass spectrometry (RDE-GC-MS). Measurements were carried out using the same rotation rate to identify the gaseous products resulting from the reduction of acetaldehyde on Cu under a well-defined mass transfer regime. Then, we employ in situ attenuated total reflection surface-enhanced infrared absorption spectroscopy (ATR-SEIRAS) and in-situ Raman spectroscopy to identify the adsorbed intermediates during acetaldehyde reduction on Cu. The potential-dependent adsorption behavior of vinyl alcohol on Cu obtained from in-situ spectroscopic results was compared to calculations using the grand-canonical density functional theory (GC-DFT). From canonical DFT modeling, we propose a detailed electroreduction pathway for vinyl alcohol to ethylene and ethane on Cu and show that the potential rate-determining step is more likely an electron transfer step rather than a commonly assumed thermochemical step or concerted proton-coupled electron transfer (CPET) step. This observation can also be generalized for the electroreduction of enols to their corresponding alkenes and alkanes in aqueous media.

## Results

### Rapid detection of products during acetaldehyde reduction on polycrystalline copper electrode

Acetaldehyde (100 mM) in pH 4, 7, and 10 is electrochemically reduced at constant potentials on a Cu disk rotating at 1600 rpm while cyclic voltammograms (CV) are collected concurrently on the Pt ring. Figure 1a–c shows the CVs measured on the Pt ring in a $N_2$ saturated $NaClO_4$ electrolyte solution at pH = 4 (0.0001 M $HClO_4$ + 0.0999 M $NaClO_4$), 7 (0.1 M $NaClO_4$), and 10 (0.0001 M NaOH + 0.0999 M $NaClO_4$), respectively, while the corresponding total disk currents are shown in Fig. 1d–f. This study focuses on a pH range between 4 and 10, considering acetaldehyde is unstable in very basic conditions (pH ≥ 12) as shown in Supplementary Fig. 9, and the Cu disk may dissolve in very acidic conditions.

The electrooxidation of acetaldehyde was first measured at each pH, referred to as the blank in Fig. 1a–c on the Pt ring electrode. The voltammogram of acetaldehyde resembles typical electrooxidation of small organic molecules on Pt[13,14], where the Pt surface is blocked by the dissociative adsorption of acetaldehyde, as indicated by the weak activity at low potentials before the adsorbates are oxidized at higher potentials (i.e., ≥0.9 V at pH = 7). The primary oxidation products are carbon dioxide and acetate in neutral and alkaline solution[15]. Oxidation features, other than those from acetaldehyde oxidation emerge on the ring detector, as a series of reduction potential is applied to the Cu disk, Fig. 1a–c. The electrochemical activity on the Pt ring that appears around 0.3 V on the CV can be attributed to the oxidation of chemical species formed during acetaldehyde reduction on Cu (i.e., products of acetaldehyde reduction). It is interesting to note that when the Cu catalyst (disk) is replaced by a Pt catalyst (disk), this oxidation activity on the Pt ring detector at 0.3 V is not observed (see Supplementary

Fig. 6), which indicates that the acetaldehyde electroreduction products on Cu are different from those on Pt. This result is supported by a recent study that ketones and aldehydes tend to adsorb dissociatively as carbon monoxide on Pt which leads to surface poisoning, blocking the Pt surface active sites from further reduction[16]. Within the potential window investigated on Pt disk (−0.2 to −0.7 V vs. RHE at pH 7), only hydrogen oxidation reaction (HOR) limiting current is observed on the Pt ring detector.

As the reduction overpotential held on Cu is increased, the "new" oxidation peaks detected on the ring increased, as expected, as more reduction products are formed at a higher reduction rate at the higher reduction overpotential. The products detected on the ring also suggest that the onset potential of this product is at −0.90, −0.75, and −0.50 V vs. RHE at pH 4, 7, and 10, respectively. On the SHE scale, the corresponding potentials are −1.14, −1.16, and −1.09 V, respectively. It is important to note that similar onset potentials were also observed by conducting the same measurement in a 0.1 M phosphate buffer solution (pH = 4, 7, and 10), shown in Supplementary Fig. 4. These findings suggest that the rate-determining step (RDS) and the step before the RDS for the formation of these reduction products on Cu may not involve proton transfer[17]. Instead, the RDS can either be a thermochemical step, water-mediated CPET step or a decoupled electron transfer step. Mechanistic arguments regarding the RDS will be discussed in more detail with DFT calculations.

The primary direct electroreduction product of acetaldehyde is ethanol over Cu[7–9], while a ethanol can also be formed from a Cannizzaro-type disproportionation reaction in a locally alkaline environment[18]. However, in the presence of acetaldehyde, ethanol oxidation does not contribute to detectable oxidation activity on the ring detector, as shown in the blank ring CVs in Supplementary Fig. 3. In this work, the dissociative adsorption of acetaldehyde is much more favorable than that of ethanol on Pt[13,15,19], and considering the relative small ethanol to acetaldehyde ratio in solution, ethanol oxidation is not detected on the Pt ring. As reduction potential of −0.5 to −0.8 V vs. RHE is applied to Cu at pH 7, negligible HOR limiting current is observed on the Pt ring detector which could be due to adsorbates displacing or inhibiting hydrogen adsorption on Cu. The adsorbates were identified using in-situ Raman and IR spectroscopy and will be discussed in more detail later in the corresponding spectroscopy section in this manuscript. From the products detected on the ring detector, we can infer that the reduction of acetaldehyde on Cu in aqueous media must be forming products other than the commonly observed ethanol and hydrogen, at relatively high overpotential (more reducing than −0.90, −0.75, and −0.50 V vs. RHE at pH 4, 7, and 10) suggesting that this reduction process may proceed through a thermodynamically unstable intermediate on Cu.

The reduction product that oxidizes at 0.3 V vs. RHE on the Pt ring detector is not detected using conventional analytical approaches such as a liquid chromatography (LC) or nuclear magnetic resonance (NMR) spectrometry[6], which highlights the superior detection sensitivity of an electrochemical ring detector. One drawback of using RRDE for product detection is that the ring electrode has limited chemical identification ability other than the oxidation features. Also, these electrochemical signatures can be rather complex when there are concurrent electroactive species formed[20].

### Coupling rotating disk electrode with gas chromatography–mass spectrometry (RDE-GC-MS) for product identification

To identify the products generated from the electroreduction of acetaldehyde on Cu, a rotating disk electrode (RDE) is coupled to an online GC-MS using an in-house built electrochemical cell filled with an $N_2$ saturated 0.1 M $NaClO_4$ solution containing 100 mM acetaldehyde. It is important to note that our online GC-MS results are qualitative rather than quantitative due to difficulties in sealing the

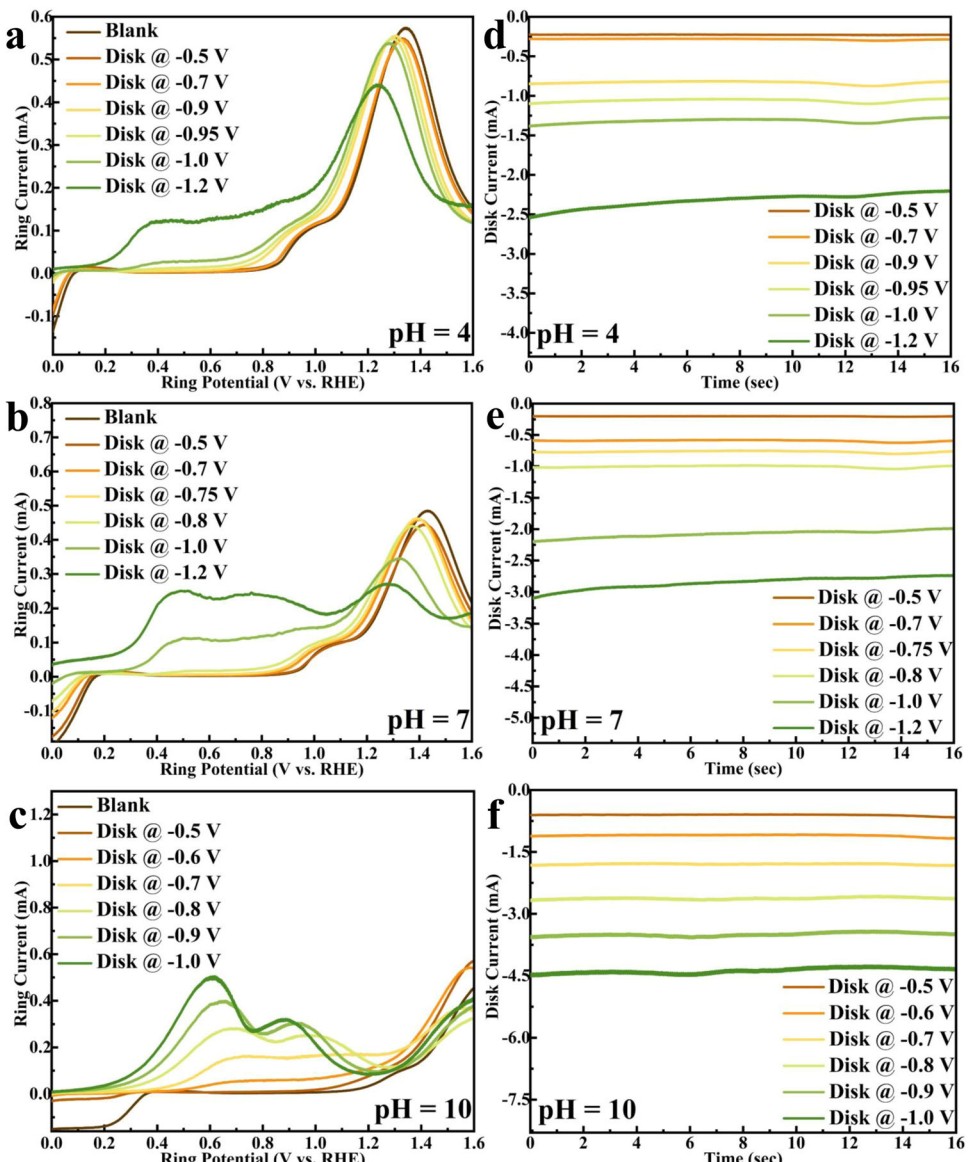

**Fig. 1 | Rapid product detection by rotating ring-disk electrode (RRDE) during acetaldehyde reduction on Cu. a–c** Cyclic voltammograms (CVs) of products detected by the Pt ring electrode during Cu catalyzed acetaldehyde reduction at pH = 4 (0.0001M HClO$_4$ + 0.0999 M NaClO$_4$), 7 (0.1 M NaClO$_4$), 10 (0.0001 M NaOH + 0.0999 M NaClO$_4$), respectively. Only the positive-going scans were shown for clarity. Cu disk potential was held from −0.5 V to −1.2 V. CV of the acetaldehyde blank was collected on the Pt ring when Cu disk was at open circuit potential (OCP). **d–f** Total Cu disk current during acetaldehyde reduction. CV conditions: 0 to 1.6 V and then back to 0 V (vs. RHE), 100 mV/s, 1600 RPM, 298 K.

electrochemical cell from the highly volatile acetaldehyde[7] and gaseous products generated.

The RDE-GC-MS system was able to directly identify the gaseous products from the electroreduction of acetaldehyde on a Cu disk under a well-defined mass transfer condition, which is necessary for understanding the reaction mechanism and for comparing experimental results across different groups[21]. At the end of the RDE-GC-MS experiment, the liquid products were collected and analyzed using NMR spectroscopy (Supplementary Fig. 10). Figure 2a shows that the gaseous products detected by GC-MS are ethylene and ethane when a reducing potential of −1.2 V vs. RHE is held on the Cu disk at 1600 RPM. This suggests that the oxidation activity observed on the Pt ring detector in the RRDE experiments is due to the electrooxidation of ethylene and ethane on Pt. The broad anodic oxidation region with two oxidation peaks ranging from ca. 0.35 to 1.1 V vs. RHE shown in Fig. 1a–c is assigned to the electrooxidation of ethylene and ethane on polycrystalline Pt. It was reported that both ethylene[22,23] and ethane[24,25]

exhibits a broad oxidation potential window with two primary oxidation peaks starting from ca. 0.4 to 1.5 V vs. RHE on Pt in acidic media. Also, the position of the oxidation peaks can shift as a function of the partial pressure of ethylene or ethane. Thus, in this work, it is not possible for us to deconvolute the ethane or ethylene partial oxidation current from the total oxidation current observed. Additionally, the overlapping ethane and ethylene oxidation make it almost impossible to assign the fundamental chemical reaction associated to the corresponding oxidation peaks. Nevertheless, the onset potential and the oxidation feature observed on the Pt ring detector is consistent with the RDE-GC-MS result.

To further explore the role of mass transfer in influencing the reaction pathway, we decreased the disk rotation rate to 900 RPM and carried out GC-MS measurements under the same condition, as shown in Fig. 2b. Signals of both ethylene and ethane decreased significantly at the lower rotation rate and completely disappeared when no rotation was applied to the electrode (Fig. 2c). The applied potential is

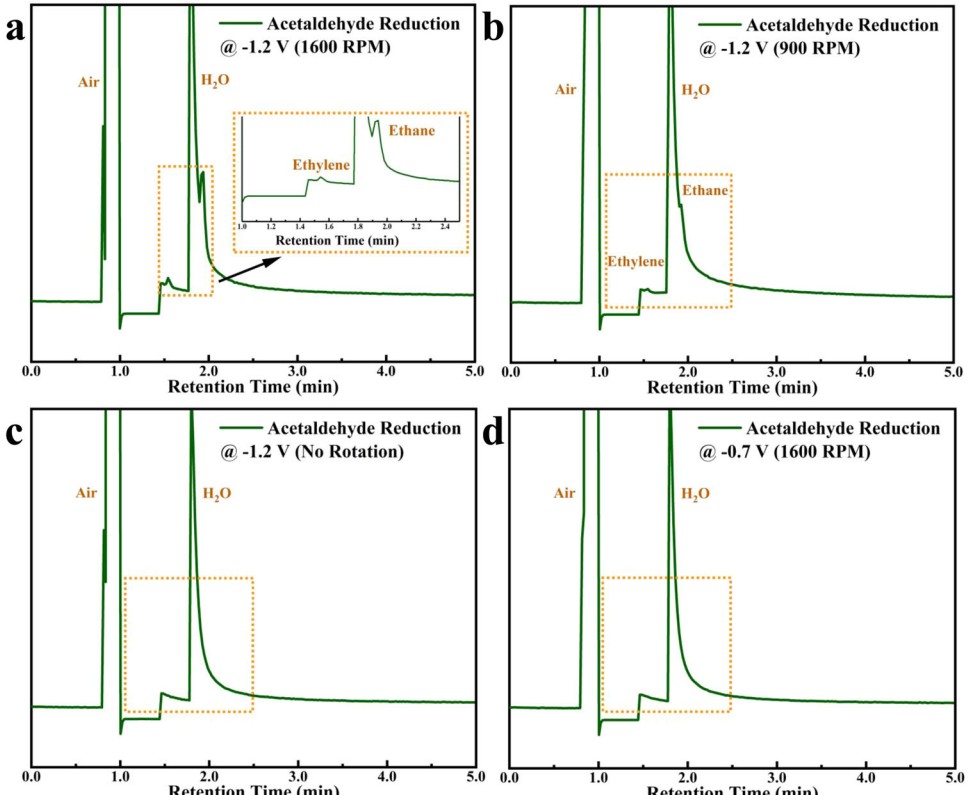

**Fig. 2 | Combined rotating disk electrode with gas chromatography-mass spectrometry measurements for detecting gaseous products generated from the electroreduction of acetaldehyde.** Gas products were auto-injected via a gas sampling mechanism and analyzed using GC-MS in selected ion monitoring (SIM) mode at: **a** −1.2 V vs. RHE, 1600 RPM; **b** −1.2 V vs. RHE, 900 RPM; **c** −1.2 V vs. RHE, no rotation; **d** −0.7 V vs. RHE, 1600 RPM in 0.1 M NaClO4 solution saturated with $N_2$ (pH = 7).

another key variable concerning the formation of ethylene and ethane from acetaldehyde. No detectable amount of ethylene and ethane was observed on the GC-MS when the Cu disk potential is held at −0.7 V at 1600 RPM (Fig. 2d), which is consistent with our RRDE measurement where the onset potential of the ethylene and ethane formation is −0.75 V on Cu (Fig. 1b).

The concurrent formation and absence of ethylene and ethane signals, depending on the reaction condition, observed from our RDE-GC-MS indicate that ethylene and ethane may share the same precursor in the electroreduction pathway. These results also confirm that the reduction pathway toward ethylene and ethane can only proceed at relatively high overpotentials (i.e., <−0.75 V at pH=7), and at high mass transfer rates during the electroreduction process. The sensitivity of the product formation to the mass transfer condition is unexpected due to the high concentration of aldehyde (100 mM) used compared to previous studies[7,9,11]. At this concentration range, along with the high reduction overpotential applied, one would expect that if ethane and ethylene are the direct products of acetaldehyde reduction that they can be easily detectable without the aid of additional convection. It is evident from the experiment conducted that the formation of these hydrocarbons are extremely sensitive to mass transport conditions. We, therefore, hypothesize that it is likely that ethylene and ethane are not the direct reduction products of acetaldehyde, but are the products of the reduction of vinyl alcohol, an enol tautomer of acetaldehyde. Acetaldehyde and vinyl alcohol are at equilibrium in aqueous solution. Considering the pKe of 6.23 for this tautomerization, the concentration of vinyl alcohol will be extremely dilute. Dilute vinyl alcohol will be very sensitive to the mass transport condition. If the dilute enol is the active reactant, its overall electrocatalytic reactivity should be under mass transport control at high electrode potentials (−1.2 V vs RHE in this work). This is consistent with our

observations that the amount of ethylene and ethane produced increased with increasing mass transport rate, while no ethylene or ethane is formed under static conditions. Our assumption and argument are further supported by in-situ IR and Raman spectroscopy combined with DFT computations. Additional experiments conducted on higher-order aldehyde under the same conditions also support this hypothesis.

In-situ spectroscopic studies were carried out to gain further insight into the mechanism of ethylene and ethane formation from the electroreduction of acetaldehyde on Cu. Figure 3a and b shows Raman spectra obtained on polycrystalline Cu in 0.1 M NaClO₄ containing 100 mM of acetaldehyde. The absence of noticeable band at 0.3 V vs. RHE indicates that there is no significant amount of adsorbate present on the Cu surface. As the potential is stepped to 0.07 V, two bands at 2920 cm⁻¹ and 2856 cm⁻¹ appeared (Fig. 3a), which can be assigned to the symmetrical stretching vibrations of the methyl group (-CH₃) and methylene group (-CH₂) from the adsorbed ethoxy intermediate (CH₃CH₂O*), respectively[10,26]. This observation is consistent with previous experimental and computational study[8,9] that the adsorbed ethoxy is the most stable adsorbed intermediate on Cu during the electroreduction of acetaldehyde to ethanol, which also shows that this reduction process is thermodynamically favorable even at 0 V vs. RHE in the absence of surface absorbed hydrogen. Literature has reported that Raman bands located between 2800 cm⁻¹ and 3000 cm⁻¹ may be from adsorbed exogenous species[27]. It is, therefore, necessary to use correlated vibrational modes located in lower frequency region and conduct the same measurement in D₂O-based electrolyte (as shown in Supplementary Fig. 21) to further support this Raman band assignment.

Another band appears at 3045 cm⁻¹, as the potential is stepped to 0.04 V (Fig. 3a), which is assigned to the stretching vibrational mode of

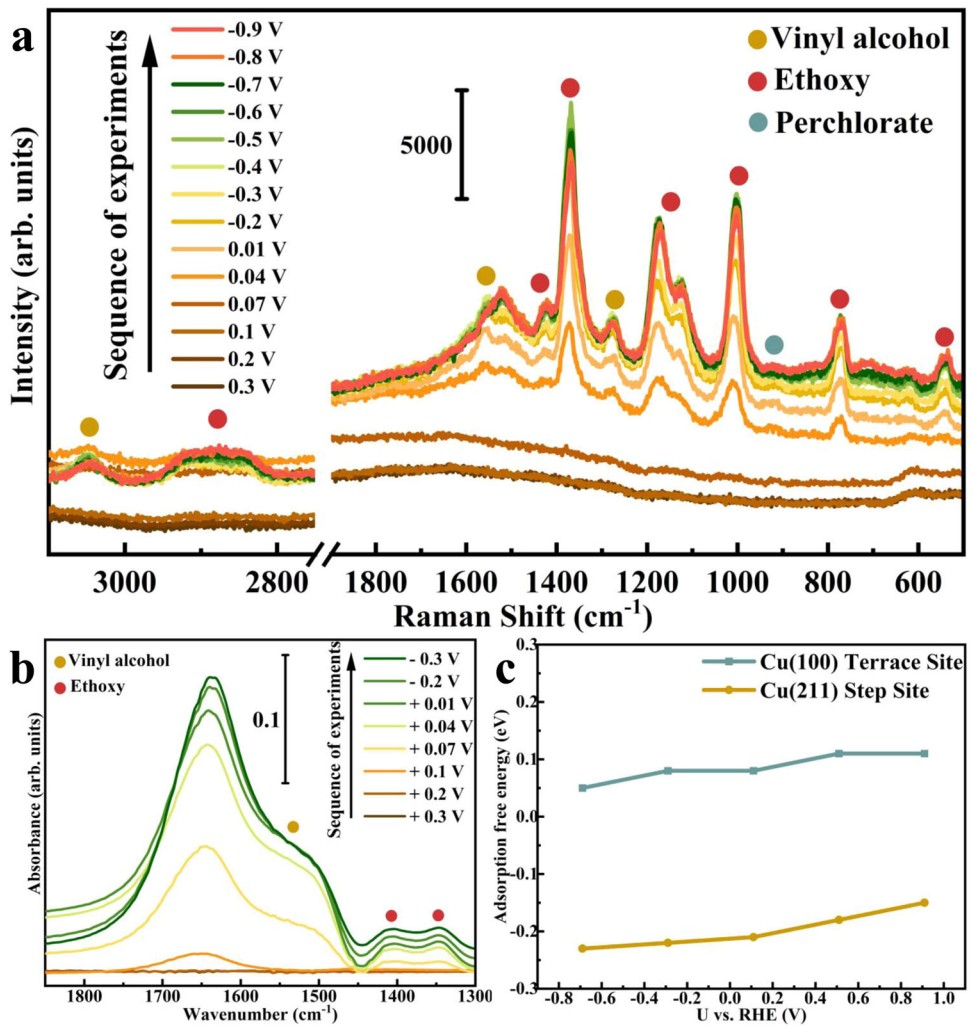

**Fig. 3 | In situ Raman and ATR-SEIRAS spectra and GC-DFT calculated results. a** In-situ Raman and **b** ATR-SEIRAS spectra (background spectrum was recorded at 0.4 V) obtained on polycrystalline copper powder electrode as a function of potential in Ar saturated 0.1 M NaClO4 (pH = 7) with 100 mM acetaldehyde. **c** GC-DFT calculated potential dependent adsorption free energy of vinyl alcohol on Cu(100) and Cu(211) as a function of potential.

methylene group (=CH$_2$) from a π-bonded intermediate[28–30]. It is interesting to note that only vinyl alcohol has a carbon double bond (C=C) connected to methylene group in our electrolyte solution. Therefore, this feature can only be assigned to the adsorption of vinyl alcohol on Cu surface. Furthermore, it is observed that this band tends to decrease as potential becomes more negative than −0.6 V, which is in agreement with our RRDE measured onset potential (ca. −0.7 V) for reducing vinyl alcohol to ethylene and ethane on Cu powder electrode as shown in Supplementary Fig. 5. The presence of a C=C is also observed at 1558 cm$^{-1}$ and 1520 cm$^{-1}$ at 0.04 V in Fig. 3b, assigned to either stretching vibration of C=C[28] or stretching vibration of a carbon-oxygen double bond (C=O) of the adsorbate as suggested by Bondue et al.[31]. Here, we exclude assigning these bands to C=O stretching mode based on previous study about acetaldehyde adsorption on Cu, which does not bond via a η$^1$(O)-CH$_3$CHO nor η$^2$(C,O)-CH$_3$CHO state in aqueous solution[7,8,11,28,32]. As such, the two adjacent bands at 1558 cm$^{-1}$ and 1520 cm$^{-1}$ are assigned to C=C stretching mode of vinyl alcohol adsorbed on two different adsorption sites on polycrystalline Cu[31]. Moreover, as shown in the ATR-SEIRAS spectra in Fig. 3c, the C=C stretching signal at 1522 cm$^{-1}$ is also present at 0.07 V. The broad signal is due to overlap with background resulting from the water bending vibrational mode. Both Raman and ATR-SEIRAS spectra support the possible adsorption of vinyl alcohol, and that this adsorption is enhanced at more negative electrode potentials.

The potential dependent adsorption free energy of vinyl alcohol is calculated on Cu(100) and Cu(211) as shown in Fig. 3d, to model the terrace site and step sites of Cu, respectively. The calculated adsorption energy trend shows that the adsorption of vinyl alcohol will become more thermodynamically favorable on both Cu(100) and Cu(211) due to the accumulation of surface electronic charge, as the potential becomes more negative. However, the step site should still be more favorable for adsorption, since the adsorption of vinyl alcohol is much more exergonic on Cu(211). This argument can be further examined using single crystal studies in the future. It is important to note that we ignored the potential dependent interaction of water with the Cu surface, therefore, our computational results may not provide an accurate estimation of adsorption energetics of vinyl alcohol at each electrode potential. Ab initio molecular dynamics simulations of cations in contact with the Cu surfaces with explicit water molecules would be needed to accurately model the possible potential dependent water displacement reaction, which is beyond the scope of this study. With this in mind, the remaining in-situ Raman and ATR-SERIAS spectra in Fig. 3b and c are assigned. Bands at 1424 cm$^{-1}$ and 1373 cm$^{-1}$ in SERIAS spectra are assigned to the asymmetrical and symmetrical bending vibrations of -CH$_3$ group from adsorbed ethoxy[10,26], respectively. This assignment also applies to the two bands at 1430 cm$^{-1}$ and 1370 cm$^{-1}$ observed in the Raman spectra. The Raman band at 1280 cm$^{-1}$ is assigned to the scissoring vibration of =CH$_2$ in vinyl alcohol[28]. A broad

band at 1150 cm$^{-1}$ can be assigned to the stretching C-C vibration of ethoxy adsorbed at different sites on polycrystalline copper[26]. A sharp band around 1006 cm$^{-1}$ can be assigned to C-O stretching of adsorbed ethoxy intermediate[10]. More importantly, we observe that C-O vibration band shifts to a lower frequency as the applied potential becomes more negative due to the Stark tuning effect as shown in Supplementary Fig. 22, consistent with a previous study reported by Chang et al. that a red-shift of C-O vibration was observed on OD-Cu electrode with more negative bias[11]. A relatively small band at 931 cm$^{-1}$ can be assigned to Cl-O stretching from perchlorate in the electrolyte solution[33], and another sharp band located at 770 cm$^{-1}$ can be assigned to the -CH$_3$ rocking vibration of ethoxy[26]. It is well established that Raman bands around 570–630 cm$^{-1}$ can be assigned to the surface adsorbed oxygen species on Cu[34,35], so it is not unexpected that the band located at 607 cm$^{-1}$ tends to diminish as potential becomes more negative due to the reduction of surface oxygen. However, the positive band at 541 cm$^{-1}$ tends to increase as potential becomes more negative so we assign this band to the Cu-O stretching vibration from the adsorbed ethoxy intermediate[36]. According to the discussion above, we can conclude that there are at least two adsorbates: vinyl alcohol and ethoxy intermediate present on Cu surface during the reduction of acetaldehyde. These species will be correlated to the electroreduction products, ethylene, ethane, and ethanol, as discussed with details along with DFT considerations.

Canonical DFT calculations were used to determine the most plausible reaction mechanism involved in the formation of ethylene and ethane. First, we focus on identifying the precursor species for the concurrent formation of ethylene and ethane as observed in our experiment. The primary product from direct acetaldehyde reduction in aqueous solution is ethanol which proceeds through adsorbed ethoxy intermediates on Cu, no ethylene or ethane was detected even at very high overpotential of −1.5 V vs. RHE, and at high concentrations of acetaldehyde (100 mM) in the electrolyte according to our previous study and experimental results reported by other groups[7–9,11]. Thus, we conclude that acetaldehyde is not likely to be the precursor species to ethylene and ethane during the electroreduction process under mass transfer control observed in this work.

Here, we hypothesize that the production of ethylene and ethane is from the reduction of surface adsorbed vinyl alcohol, which is the enol tautomer of acetaldehyde. The keto-enol equilibrium constant (pK$_e$ = 6.23) suggests that the enol tautomer concentration in the bulk is very low[5], while surface adsorbed species suggest the presence of the vinyl alcohol. We then modeled vinyl alcohol reduction on two representative Cu surface sites: terrace site on Cu(100) and step site on Cu(211), where the C-O bond of vinyl alcohol must be cleaved first before it can be hydrogenated to ethylene and ethane. We assume that C-O bond scission of adsorbed vinyl alcohol is the RDS.

One possible reaction pathway for C-O bond scission is a direct hydrogen addition to the hydroxyl group of vinyl alcohol to form *CHCH$_2$ and water, which follows a CPET mechanism, in which water acts as proton donor. However, the activation barrier of this direct electrochemical hydrogenation step is reported to be around 1.5 eV at 0 V vs. RHE on Cu(100)[37]. This suggests that the direct hydrogenation of hydroxyl group is considered to be kinetically blocked under our experimental condition, as the threshold activation barrier for reasonable kinetics is generally thought to be lower than 0.75 eV at room temperature[38]. Some literature report indicates water-mediated reaction may be pH independent, however, this is not consistent with most experiments, such as those on gold or copper in alkaline and neutral media[39–41]. Extending this rationale, if water reduction reaction is pH dependent, then the RDS cannot be a water-mediated CPET step. Furthermore, our RRDE carried out in acidic (pH = 4), neutral (pH = 7), and basic(pH = 10) solution under a well-defined high mass transport condition, all measured onset potentials that are independent of pH on the SHE scale. Therefore, it is unlikely

that the RDS involves C-O bond of vinyl alcohol being cleaved by a water-mediated CPET step.

The second possible pathway is a thermochemical step for C-O bond scission that occurs on Pt electrode[31,42]. However, our calculated activation barriers for C-O bond scission on Cu(100) terrace site and Cu(211) step site are 1.30 eV and 1.14 eV, as shown in Fig. 4a, b, respectively. These barriers are too high, as such, thermochemical C-O bond scission is also regarded as kinetically hindered on Cu surface. It is interesting to note that the activation barrier of C-O bond scission on Pt(100) terrace site is much lower (0.99 eV, as shown in Fig. 4c), which is consistent with previous study[42] that reported terrace site of Pt(100) being effective for C-O bond breaking. It is important to note that pre-exponential factor and explicit solvent effect are not included in these computations, thus, these computational results must be compared with experimental results to draw any reliable conclusion. While calculation predicts that C-O bond scission of vinyl alcohol is much more kinetically favored on Pt, and thus, may be able to facilitate ethylene and ethane formation, this barrier is still too high. Also, ethylene and ethane formation from the reduction of acetaldehyde on Pt is not observed experimentally using our RRDE under the same reaction conditions (See Supplementary Fig. 6). Furthermore, according to the literature[6,43,44], several compounds along the CO$_2$ reduction pathways, such as glyoxal (OHC-CHO), glycolaldehyde (OHC-CH$_2$OH), and ethylene glycol CH$_2$OH-CH$_2$OH), which contains C-O bond, were not precursors to the formation of ethylene (or ethane) in CO$_2$ reduction on Cu. These experimental evidence support that thermochemical C-O bond scission may not be favorable on Cu in aqueous media at room temperature. Otherwise, it is difficult to explain why these products are not further reduced to alkanes and alkenes at high reducing electrode potentials. As a result, we conclude that a thermochemical C-O bond scission is unlikely to happen on Cu surface under our experimental condition.

Thus, the remaining pathway for breaking the C-O bond is a decoupled proton-electron transfer step. Although most proton-coupled electron transfer processes are considered to be CPET to avoid high-energy charged intermediates in heterogeneous catalysis[45], recent experimental evidence points to adsorbed charged intermediate stabilized on the electrode surface through a short-range electrostatic interaction due to the presence of cations near the double layer region[46]. Moreover, CO dimerization was believed to proceed via a decoupled electron transfer to the adsorbed CO molecules to form C$_2$O$_2^-$ intermediates on Cu, and proton transfer only happens after the formation of this negatively charged intermediate[38,47,48]. Hence, we propose that the C-O bond scission of vinyl alcohol is induced by a decoupled electron transfer step followed by a β-elimination process as shown in Fig. 4d after ruling out another plausible mechanism. This hypothesis is consistent with all our measurements, including the RRDE data that show the onset of vinyl alcohol to ethylene and ethane to be independent of pH (ca. −1.1 V vs. SHE regardless of solution pH), where the electron transfer step is decoupled from proton transfer. A similar kinetically favorable beta-elimination step is also reported in a previous full atomistic computational modeling on the CO reduction to ethylene at Cu(111)[49].

We further propose that the first electron should be transferred to the beta carbon (C$_β$) of the vinyl alcohol by taking into account the fact that the hydroxyl group (·OH) is an electron donating group in the keto-enol equilibrium so that the charged enol intermediate will be destabilized if the electron is transferred to alpha carbon (C$_α$) of the vinyl alcohol. After the first electron transfer step, C-O bond is cleaved by beta-elimination after the addition of one proton to the charged intermediate. According to DFT calculations (see Supplementary Fig. 29), another argument used to support that electron will be transferred to C$_β$ instead of C$_α$ is that the adsorbed vinyl alcohol with a charged C$_α$ will result in an unstable *CH$_2$CH intermediate after hydroxyl group leaves as water, upon the addition of one proton,

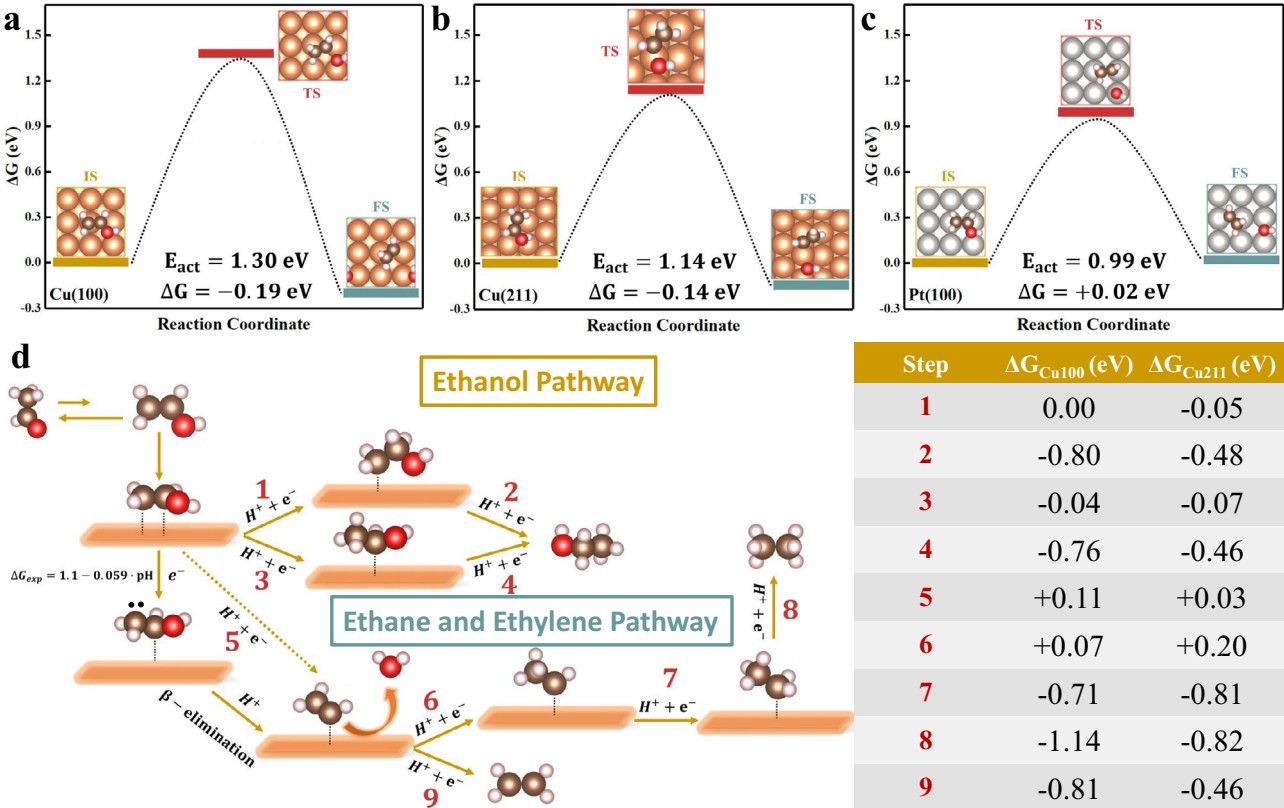

**Fig. 4 | Proposed reaction pathways for electrocatalytic reduction of vinyl alcohol on Cu. a–c** Free energies of the initial (IS), transition (TS), and final states (FS) for the C-O bond-breaking processes of adsorbed vinyl alcohol on Cu(100), Cu(211), Pt(100), respectively. $E_{act}$ values correspond to the calculated NEB barriers with ZPE correction. **d** Proposed reaction pathway for vinyl alcohol reduction to ethylene, ethane, and ethanol on Cu(100) and Cu(211). All $\Delta G$ values are given at 0 V vs. RHE based on CHE model. Solid arrows indicate proposed pathway and dashed arrow denotes unlikely pathway. Color code: hydrogen in white, oxygen in red, carbon in brown, copper in orange, platinum in gray.

which is much more thermodynamically unfavorable than the ˙CHCH₂ intermediate on Cu. In addition, the onset potential of −1.1 V vs. SHE can be regarded as an approximation of the equilibrium potential for the decoupled electron transfer step if we assume very fast electrokinetics for the decoupled electron transfer process after the thermodynamic barrier is overcome[45]. We use this approximation to estimate the free energy change for the decoupled electron transfer step, which can be expressed as $\Delta G = 1.1 - 0.059 \cdot pH$ on the RHE scale by assuming a typical Nernstian shift at 298 K. Consequently, a relatively high electrode potential is required to overcome the thermodynamic barrier of the first electron transfer step, especially in acidic and neutral media. Once the charged intermediate is formed under a relatively high electrode potential, all the other steps from vinyl alcohol to ethylene and ethane are highly exergonic as shown in Fig. 4d. Furthermore, as shown in Supplementary Fig. 28, our Bader charge analysis shows that $C_\beta$ has a deficient electronic charge of 0.52 e⁻ in adsorbed vinyl alcohol on Cu(211), whereas the $C_\beta$ has an excess electronic charge of 0.14 e⁻ on Cu(100). As such, the $C_\beta$ of vinyl alcohol will more readily accept one electron from the electrode on the Cu(211) step site based on this simple electrostatic argument, which can be further examined by future single crystal study.

Most previous studies proposed that the adsorbed vinyl alcohol was the direct precursor to ethanol during CO₂ reduction on Cu[6,37,48]. In this work, we cannot exclude the possibility that vinyl alcohol can be reduced to ethanol under our experimental condition since ethanol was detected in ¹H NMR spectrum as shown in Supplementary Fig. 10 and nearly all the steps towards ethanol formation from vinyl alcohol are thermodynamically downhill at 0 V vs. RHE on Cu(100) and Cu(211). However, according to our experimental results, we conclude that vinyl alcohol can also be reduced to ethylene and ethane at high

overpotentials on Cu, which will compete with vinyl alcohol reduction to ethanol. This interesting result provides additional insights into the pathways that can lead to ethane and ethylene during CO₂ reduction on polycrystalline Cu electrode[50].

## Extension of proposed mechanism to higher order ketone and aldehyde

Similar RRDE and RDE-GC-MS experiments were carried out with propionaldehyde and acetone reduction under the same experimental condition on Cu, which are presented in Supplementary Figs. 7, 8, 11, 12, 14 and 15. These aldehydes and ketone are chosen according to their larger $pK_e$ of keto-enol tautomerization equilibrium and solubility in aqueous solution. Summarized results from this work and other reports are shown in Table 1:

Under our experimental conditions, we assume that tautomerization is not catalyzed by Cu due to the weak adsorption of acetaldehyde on Cu under typical electroreduction condition[7,8], and that alkanes and alkenes are not detected under static condition. There are two key factors that determine the reaction activity of enol reduction on Cu: First, the $pK_e$ value of the keto-enol tautomerization should not be too large. Otherwise, the concentration of the enol will be too low resulting in the low production rate of deoxygenated products from the enol reduction. Here we recommend using the $pK_e$ value of acetaldehyde as a benchmark to compare enol reduction of other aldehydes and ketones in aqueous solution. For example, propene and propane signals are more prominent and are easily detected in the GC-MS from the propenol reduction in a 100 mM propionaldehyde in 0.1 M NaClO₄ solution, due to their smaller $pK_e$ value than that of acetaldehyde. However, no propane or propene signal are detected in 100 mM acetone in 0.1 M NaClO₄ solution. Compared to acetaldehyde,

**Table 1 | Summary of RDE-GC-MS, RRDE results, and $pK_e$ values for selected aldehydes and ketone (−1.2 V vs. RHE on Cu disk, 1600 RPM, 0.1 M $NaClO_4$, 298 K)**

| Aldehydes/ketone | $pK_e$ of keto-enol tautomerization | Detected products from enol reduction | Onset potential (V vs. SHE) | Reference |
|---|---|---|---|---|
| Acetaldehyde | 6.23 | Ethylene and ethane | ~ −1.1 | 5 |
| Propionaldehyde | 5.10 | Propene and propane | ~ −0.7 | 62 |
| Acetone | 8.33 | Not detected | N/A | 63 |

acetone has a much larger $pK_e$ value due to the stabilization of the keto form via electron donation to the positively charged carbonyl carbon atom from the two alkyl groups[5] which results in a much lower enol concentration in the bulk solution.

Second, a relatively high reducing electrode potential must be applied to facilitate the first electron transfer and subsequent enol reduction on Cu. Compared to vinyl alcohol, propenol requires a lower onset potential of −0.7 V vs. SHE to produce its corresponding alkanes and alkenes in our experiment. This is possibly due to the required free energy of transferring one electron to adsorbed propenol is smaller than that to the adsorbed vinyl alcohol on Cu surface. In theory, a detailed evaluation of the energetics of a charged intermediate adsorbed on electrode surface requires maintaining a constant electrode potential in the electronic structure calculations, and in modeling complex interactions at the electrochemical double layer[51], which is beyond the discussion of this work. Therefore, once the onset potential for the first electron transfer is reached, we conclude that the enols from soluble aldehydes and ketones can also be reduced to their corresponding alkanes and alkenes on polycrystalline Cu electrode in aqueous electrolyte by following a similar reduction mechanism.

## Discussion

New insight has been provided on the electrochemical reduction of enols on Cu electrode. We employed a highly sensitive experimental methodology involving a RDE coupled to an online GC-MS, that provides a well-defined mass transport control enabling the pre-concentration of enols to the catalyst surface and the identification products that are otherwise undetected. Reported here, for the first time, our results suggest that enols can be reduced to their corresponding alkanes and alkenes at high electrode potentials on Cu surface.

In this work, we detected the products from vinyl alcohol reduction occurring on Cu, first from the CV obtained on a Pt ring detector in an RRDE setup. These products were confirmed to be ethane and ethylene from subsequent GC-MS analysis. The onset potentials for the formation of ethane and ethylene were determined at pH 4, 7, and 10 and were found to be pH independent on the SHE scale. In-situ Raman and ATR-SEIRAS results show that two adsorbates: vinyl alcohol and ethoxy intermediate are present on the Cu surface during the reduction process. Adsorbed vinyl alcohols tend to accumulate on Cu as the potential is stepped to more negative values. Observations from the spectroscopic studies are supported by GC-DFT calculations.

The combined experimental work with DFT modeling allows us to conclude that electrocatalytic reduction of soluble enols will give rise to their corresponding alkenes and alkanes on Cu. Moreover, the RDS step is more likely to be an electron transfer step rather than commonly assumed thermochemical step or CPET step. The reaction rate of enol reduction highly depends on its stability ($pK_e$) in solution.

In conclusion, the results of this study provide new insight into the electrochemistry of enols in aqueous electrolyte by (1) identifying previously unreported products (alkanes and alkenes) from enol reduction, (2) deducing reaction mechanisms for the electrocatalytic reduction of enols and (3) extending these findings to predict the reactivity of higher order soluble ketone and aldehyde. The electrochemical pathways revealed from the enol to alkane and alkene conversion not only enriches our understanding of the electrochemical conversion pathways for $CO_2$ toward $C_2$ and $C_3$ products, it also potentially opens strategies for a direct electrosynthesis of terminal olefins from aldehydes in ambient, aqueous conditions. Moreover, this work has implications for a potentially economically viable way to deoxygenate and refine bio-oil through an electrochemical deoxygenation method.

The conclusions drawn from this work show that mass transport effect is a key factor in probing the reduction activity of enols on Cu in aqueous media. Thus, a detailed evaluation of kinetic model is required to elucidate the reaction activity of enols. It is important to note that there are four possible kinetic models that can account for the formation of ethylene and ethane based on the interplay between mass transport rate and keto-enol tautomerization rate in the solution. If tautomerization is slower than the mass transport of acetaldehyde or vinyl alcohol to the electrode surface, two models can be built according to (1) acetaldehyde as the precursor and (2) vinyl alcohol as the precursor. If tautomerization is faster than the mass transport of acetaldehyde or vinyl alcohol to the electrode surface, another two models can be built based on (3) acetaldehyde as the precursor and (4) vinyl alcohol as the precursor. Under the mass transport limiting condition, only direct precursor contributes to the mass transport limiting current in models (1) and (2) due to a much slower tautomerization process. However, both acetaldehyde and vinyl alcohol will contribute to the mass transport limiting current in models (3) and (4) due to a much faster tautomerization process.

Thus, future work will focus on a detailed numerical simulation of mass-transport limiting current values for different models and compare them with experimental values, which will provide a more rigorous analysis for determining the kinetic model.

## Methods

### Chemicals and materials

Acetaldehyde (≥99.0%), propionaldehyde (≥99.0%), acetone (≥99.0%), isopropanol (≥99.5%), sodium perchlorate (HPLC grade), sodium phosphate monobasic (≥99.0%), sodium phosphate dibasic (≥99.0%), deuterium oxide (99.9 atom% D), copper powder (<45 μm, 99.7%, trace metals basis), and chloroauric acid ($HAuCl_4 \cdot xH_2O$, 99.99 %) were purchased from Sigma Aldrich. Sodium hydroxide (≥99.0%) and 85% phosphoric acid (ACS certified) were purchased from Fisher Scientific. Nafion dispersion (D520) was purchased from Dupont. A 1 M stock solution of $NaClO_4$ was prepared with ultrapure de-ionized water (≥18.2 MΩ cm, Millipore Milli-Q, 25 °C) and pre-electrolyzed for at least 24 h. A 0.1 M $NaClO_4$ electrolyte solution was prepared by diluting 1 M pre-electrolyzed stock solution with ultrapure water. A 0.1 M $HClO_4$ solution was diluted directly from the concentrated $HClO_4$ (Fisher Scientific, trace metal grade). Calibration gas mixture was ordered from Linde, Inc. Other gases used in this work had a purity of ≥99.998% and were used as received from Praxair, Inc.

### Disk electrodes preparation

Polycrystalline Cu (RRDE-Cu) and Pt disks (diameter: 5 mm, thickness: 4 mm, geometric surface area: 0.196 $cm^2$, mirror polished, 99.99%) used in RRDE setup were purchased directly from PINE Research Instrumentation. An in-house RDE was made by the ASC Machine Shop at The Ohio State University. The polycrystalline Cu disk (RDE-Cu) used in the in-house RDE setup has a diameter of 10 mm enclosed in a Teflon

shell. The geometric surface area of this Cu disk electrode is 0.785 cm$^2$, which is four times larger than the disk electrode used in RRDE setup. Before each use, Pt disk electrode was first mechanically polished using micro cloth and 0.05 μm alumina slurry (Buehler Company), and then electrochemically polished in 0.1 M HClO$_4$ by cycling between 0 and 1.6 V (V vs. RHE) at 1 V/s for 100 cycles until a steady-state cyclic voltammogram of Pt was observed. RRDE-Cu disk electrode was polished with 600-grit sandpaper, which was then followed by using a series of alumina slurry, in the following order of alumina particle size: 1, 0.3, 0.05 μm, on a micro cloth (Buehler Company) until mirror polished. After which, the RRDE assembly was sonicated three times (5 min each) in ultrapure water (≥18.2 MΩ cm, Millipore Milli-Q, 25 °C) and then transferred to the electrochemical cell. RDE-Cu disk electrode was polished using a micro cloth and 0.05 μm alumina slurry and sonicated three times (5 min each) in ultrapure water. Then, an electrochemical polishing procedure was followed to further clean the surface, which included holding potential of 1.8 V versus Ag/AgCl (saturated KCl) reference electrode (Biologic VSP potentiostat) on the Cu disk in 85% phosphoric acid for 180 s with a rotation rate of 900 RPM, until a mirror shiny surface was observed. the RDE assembly was again sonicated three times (5 min each) in ultrapure water and then transferred to the electrochemical cell. All parameters were kept constant for every experiment to ensure reproducibility.

### Ring electrode preparation
A rotating ring-disk electrode with an interchangeable disk was purchased directly from PINE Research Instrumentation (disk diameter: 5.0 mm; ring outer diameter: 7.5 mm; ring inner diameter: 6.5 mm; ring material: Pt with a purity of 99.99%). After the metal disk was inserted into the RRDE, the entire electrode assembly was polished using a micro cloth and 0.05 μm alumina slurry. The Pt ring electrode was further cleaned electrochemically by cycling between 0 and 1.6 V at 1 V/s for 100 cycles in 0.1 M HClO$_4$ until a steady-state cyclic voltammogram resembling that of a clean Pt surface was observed. When a Cu disk was used in the RRDE assembly, this cleaning procedure was performed in 0.1 M NaClO$_4$ due to the dissolution of Cu in 0.1 M HClO$_4$.

### RRDE experiments
Electrochemical reduction experiments were performed in a standard three-electrode cell using a rotating ring-disk electrode assembly equipped with a bipotentiostat (CH Instrument, CHI 760D). The working electrode and counter electrode compartment were separated by a fine porous frit. A reversible hydrogen electrode (RHE) with a Luggin capillary and a high surface area Pt mesh used as the reference electrode and the counter electrode, respectively. Before each experiment, N$_2$ gas was introduced into the electrolyte solution for at least 20 min to remove dissolved O$_2$. Then, cleaned RRDE assembly was transferred to the electrochemical cell. A constant negative potential was applied to the disk electrode to reduce aldehydes. Meanwhile, a cyclic voltammogram of Pt ring electrode was recorded from 0 to 1.6 V with a scan rate of 100 mV/s at 1600 RPM. The electrolyte was continuously purged with N$_2$ during the measurement. A rotation rate of 1600 RPM was found to be the optimum for transporting the gas bubbles generated on the disk away from its surface as well as to obtain a stable detection signal on the ring electrode. The solution resistance was determined by carrying out electrochemical impedance spectroscopy at open circuit potential. The electrode potential was compensated at 85% of the ohmic drop. All parameters were kept constant unless otherwise specified.

### Combined RDE with GC-MS measurements
Electrochemical measurement conditions were the same as those used in the RRDE experiments. During electroreduction experiments, the RDE setup was mounted on a rotator shaft and the rotation rate was controlled by a rotating electrode speed controller (PINE Research

Instrumentation). The gas outlet of the cell was connected to a series of two gas-sampling loops to introduce the gaseous products formed during electroreduction of aldehydes into a gas chromatograph (7890A Agilent Technology). The first gas sampling loop was connected to a quadrupole mass spectrometer through a CP-PoraBond Q capillary column. The second gas-sampling loop was connected to a thermal conductivity detector (TCD) through a Haysep Q and Molseive 5A column in series, which was used to detect hydrogen. The mass spectrometer was used to detect hydrocarbons like C$_2$H$_4$, C$_2$H$_6$, C$_3$H$_6$, and C$_3$H$_8$. Specifically, selected ion monitoring (SIM) mode was chosen to detect these hydrocarbons because SIM mode enables one to filter out mass fragments (i.e., 16, 18, 28, etc.) from the ambient background, which helps increase the detection sensitivity and achieve a lower detection limit for interested hydrocarbons. Gaseous products were auto-injected and analyzed by GC-MS every 10 min. The retention times of C$_2$H$_4$, C$_2$H$_6$, C$_3$H$_6$, and C$_3$H$_8$ were calibrated using standard gas mixtures (Linde).

### Liquid species characterization
Liquid reactants and products were analyzed using a Bruker 400 MHz NMR spectrometer. NMR samples were prepared by mixing 800 μL of the electrolyte solution containing liquid reactants and products with 100 μL of D$_2$O and 100 μL of 100 ppm acetonitrile as an internal standard. The water signal was suppressed in measured $^1$H spectrum by using the excitation sculpting method implemented in our NMR spectrometer. Each spectrum was obtained from 128 scans.

### Preparation of Cu powder electrode
First, 100 mg of commercial Cu powder was dispersed in 2.5 mL isopropanol and 30 μL of Nafion solution (5 wt%) was added into the mixture with sonification for 1 h to form a uniform catalyst ink. Then 100 μL ink was dropped onto a polished glass carbon electrode (Φ = 10 mm, Gaoss Union), and dried in N$_2$ atmosphere for the subsequent in-situ Raman tests. For in-situ ATR-SEIRAS tests, 30 μL ink was dropped on the Si crystal, and dried in N$_2$ atmosphere before test. X-ray diffraction characterization of Cu powder electrode was shown in Supplementary Fig. 30.

### In-situ Raman spectroscopy
A photograph of in-situ Raman setup (Renishaw-RL633) was shown in Supplementary Fig. 27, an electrolytic flow cell with three-electrode configuration was used for the in-situ Raman test. Cu powder electrode was used as working electrode, a graphite rod (Sigma-Aldrich, 99.999%) and Ag/AgCl (saturated KCl) electrodes were used as counter and reference electrodes, respectively. The cathodic cell and anodic cell were separated by an ion exchange membrane (117, Dupont). During Raman test, Ar-saturated 0.1 M NaClO$_4$ solution was continuously fed into the electrolytic flow cell at a rate of 15 mL/min. Raman spectra were collected in the range of 100 to 3500 cm$^{-1}$ using a 633 nm HeNe-Laser. All the potentials were converted to be on RHE scale.

### In-situ ATR-SEIRAS
The gold film was chemically deposited onto a silicon ATR crystal[52]. First, 0.105 g NaOH was dissolved in 67 mL ultrapure water and mixed with 3 mL HAuCl$_4$ (0.1 mg/mL) to obtain an orange-red solution A. Then 0.1337 g NH$_4$Cl (≥99.5%, Aladdin), 0.6025 g Na$_2$S$_2$O$_3$·5H$_2$O (≥99.999%, trace metal basis, Aladdin), and 0.9653 g Na$_2$SO$_3$ (ACS grade, Greagent) were dissolved in 30 mL ultrapure water to obtain solution B. After that, 7 mL solution A and 3 mL solution B were mixed and kept stirring for at least 3 h until the mixture became clear. The resulted solution was denoted as solution C. Before the chemical deposition of Au, the Si crystal was mechanically polished with 50 nm Al$_2$O$_3$ powder for 10 min and then soaked in piranha solution (6 mL 98% H$_2$SO$_4$ and 2 mL 30% H$_2$O$_2$) for 1 h, and finally rinsed with ultrapure

water and ethanol. The silicon surface became hydrophobic after the treatment. The surface of Si was immersed in 40% $NH_4F$ solution for 90 s, and then immersed in a mixture of 4 mL of solution C and 34 μL 40% hydrofluoric acid at 55 °C for 5 min, forming a thin and clean Au film after washing with ultrapure water and ethanol.

A schematic of ATR-SEIRAS setup consisting of two chambers with three electrodes is shown in Supplementary Fig. 26, 3 mL and 10 mL of 0.1 M $NaClO_4$ electrolyte was injected into anodic and cathodic chamber, respectively. Cu powder electrode was used as the working electrode. Pt foil was used as the counter electrode and reference electrode was the same as that used in Raman tests. Anode chamber and cathode chamber were separated by Nafion membrane to prevent the interference from Pt CE reported by Dunwell et al.[53]. No electrolyte flow was applied in our ATR-SEIRAS tests. An INVENIO R FT-IR (Bruker) spectrometer equipped with a MCT detector was used for the in-situ ATR-SEIRAS test and all spectra were given by the absorbance ($-\log(R/R_0)$). The spectral resolution was 4 cm$^{-1}$ and the scanning time was 30 s for each spectrum. All the potentials were converted to be on RHE scale.

## Computational details

The canonical DFT calculations were carried out using VASP code[54]. The exchange-correlation functional was described with generalized gradient approximation method in the version of the Perdew–Burke–Ernzerhof (PBE) functional[55]. The Cu(100), Cu(211), and Pt(100) surfaces were modeled using (3 × 3 × 4) supercells (bottom two layers fixed) with vacuum layers of 15 Å and dipole corrections were applied. All geometries were fully optimized with a plane wave energy cutoff of 415 eV and the electronic forces and energy were converged within 0.03 eV/Å and $1.0 \times 10^{-5}$ eV, respectively. The surface Brillouin zone was sampled with a 4 × 4 × 1 Monkhorst Pack k-point meshes[56]. The Fermi level of metal surfaces was smeared by the Methfessel–Paxton approach with a smearing width of 0.2 eV. Isolated molecules were calculated in boxes of 15 Å × 15 Å × 15 Å using Gaussian smearing and a smearing width of 0.001 eV with the gamma point only. The relative charges associated with individual atoms were obtained using Bader charge analysis. The DFT-calculated electronic energies were converted into free energies at 298.15 K as outlined below:

$$G = E_{DFT} + E_{ZPE} + \int_0^{298.15} C_p dT - TS \quad (1)$$

where zero-point energies ($E_{ZPE}$) of adsorbates and gas molecules were calculated from the vibrational frequencies obtained within the harmonic oscillator approximation. Only vibrational contributions to the thermodynamic corrections were considered for adsorbed species. Free energies of proton–electron pairs were calculated using the computational hydrogen electrode (CHE) model[57]. Solvation effect was modeled as an external correction depending on the chemical nature of the adsorbates, which can be seen in Supplementary Tables 2 and 3. The transition states were determined with the climbing image nudged elastic band method[58]. Seven images were generated between the initial state and the final state by a linear interpolation of the coordinates. Each transition state was verified by having only one imaginary frequency.

To consider the surface charge effect in adsorption, the grand-canonical DFT calculations were carried out using the fictitious charge particle (FCP) method[59] implemented in Quantum ESPRESSO (QE) code[60]. The PBE exchange-correlation functional[55] was used with a plane-wave basis within the ultrasoft pseudopotential framework. Kinetic energy and charge density cutoffs were set to 32 Ry and 320 Ry (1 Ry = 13.606 eV), respectively. The Cu(100) and Cu(211) slabs were modeled using the same 3 × 3 × 4 supercells. The bottom two layers were fixed throughout the structural optimization. The surface Brillouin zone was sampled using 4 × 4 × 1 Monkhorst Pack k-point

meshes[56] and the electronic occupations were smoothed using Marzari-Vanderbilt cold smearing with a smearing width of 0.01 Ry. In the FCP calculations, the convergence thresholds for force ($F_e$) and grand potential ($\Omega$) are set to $1.0 \times 10^{-2}$ eV and $1.0 \times 10^{-6}$ Ry, respectively. The applied electrode potential U on standard hydrogen electrode (SHE) scale can be correlated to the Fermi level ($E_F$) of the system:

$$U = \frac{-E_F - \phi_{SHE}}{e} \quad (2)$$

where $\phi_{SHE} = 4.44$ eV was the value of the work function of the SHE used in this work, which was taken from the experimental results reported by Trasatti[61], $E_F$ is the value of Fermi level of the system which can be fixed as a constant during the grand-canonical calculations. Thus, the adsorption free energy at constant electrode potential U was defined as:

$$G_{ads}^M(U) = G_{M^*}(U) - \left[G_M(U) + G_*(U)\right] \quad (3)$$

in which * denotes an adsorption site on the slab surface and $M^*$ is the adsorbate. The free energy of a metal slab was calculated directly from the grand-canonical DFT calculations:

$$G_*(U) = \Omega_*(U) \quad (4)$$

and the free energy of adsorbate $M^*$ was calculated as following:

$$G_{M^*}(U) = \Omega_{M^*}(U) + E_{ZPE} + \int_0^{298.15} C_p dT - TS \quad (5)$$

where the grand potential $\Omega_{M^*}(U)$ was determined from grand-canonical DFT calculations and the thermodynamic and solvation corrections were the same as those used in canonical DFT calculations. All the potentials were converted to be on RHE scale in the main article.

## Data availability

The data that support the findings of this study are available within the article and its Supplementary Information files or from the corresponding author upon request.

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

## Acknowledgements

This work was supported by the National Science Foundation (NSF CHE 1455162) and in part by the Ohio State University Materials Research Seed Grant Program, the Ohio State Department of Chemistry and Biochemistry, and the National Natural Science Foundation of China (Grant Nos. 21822601, 22176029). We acknowledge the Ohio Supercomputing Center for providing computational resources.

## Author contributions

Z.C. and X.D., collected, analyzed, visualized the data, and participated in the writing of the manuscript. Z.C. and X.D. contributed equally to this manuscript. S.G.C., W.D., and M.N.T. participated in the collection and discussion of the data. F.D. procured funds to facilitate this work and participated in the discussion of research findings. A.C.C. participated in the procurement of research funds, design, analysis, discussion, and writing of the manuscript.

## Competing interests

The authors declare no competing interests.
