## [Peer review file · Nature Communications]

REVIEWER COMMENTS

Reviewer #1 (Remarks to the Author):

The authors report the electrochemical conversion of vinyl alcohol and acetaldehyde on polycrystalline Cu to ethanol, ethylene and ethane; and propenol and propionaldehyde to propanol, propene and propane. Sensitive detection was achieved using a rotating disk electrode coupled with gas chromatography-mass spectrometry (RDE-GC-MS). In situ attenuated total reflection surface-enhanced infrared absorption spectroscopy (ATR-SEIRAS), and in situ Raman spectroscopy confirmed the adsorption of the vinyl alcohol. Calculations using canonical and grand-canonical density functional theory (DFT and GC-DFT), along with experimental findings suggest that the rate-determining step (RDS) for ethylene and ethane formation is an electron transfer step to the adsorbed vinyl alcohol. The experimental phenomena and the experimental conclusions are very interesting, and indeed innovative reaction mechanisms are proposed. Therefore, I recommend the publication in Nat. Commun. upon revisions

1. The authors used Pt ring electrodes for the identification of acetaldehyde reduction intermediates. The authors also mentioned that acetaldehyde in solution would adsorb to the Pt ring electrode surface and be oxidized to give CO₂ and acetic acid (Line 97, Page 4). It is possible that CO₂ may reabsorb on the surface of the Cu disc electrode and be reduced to obtain products such as ethylene. How did the authors avoid the reduction of CO₂ on the Cu surface during the test? Or how did the authors decouple the reduction current of CO₂ and the reduction current of acetaldehyde (enol) during the test?

2. The authors mentioned that no oxidation current of ethanol was detected in the Pt ring (Supplementary Fig. 3). Such a phenomenon is interesting and critical. With such a phenomenon, the authors suggest that the observed oxidation peaks located at 0.3 V vs. RHE and 0.9 V vs. RHE should be species other than acetaldehyde and ethanol (i.e., ethylene and ethane). However, it is still difficult for me to understand why the oxidation of ethanol is not detected. Acetaldehyde is an intermediate for ethanol production (DOI: 10.1002/anie.201508851; 10.1021/jacs.8b04058), so the concentration of ethanol should be higher than other electrolysis products. In addition, there are various reactions for the oxidation of ethylene, which may give CO₂, ethylene oxide (10.1126/science.aaz8459), ethylene glycol (DOI: 10.1038/s41929-019-0386-4) and etc. Thus, which reaction does the oxidation peak detected by the authors correspond to? Meanwhile, the thermodynamic potential of the above mentioned oxidation reaction of ethylene is not always more negative than that of ethanol (e.g., 1.0 V vs. RHE for the oxidation of ethylene to ethylene oxide). If these oxidation reactions can be detected, why is the oxidation of ethanol not detected? Is there a poisoning of the smooth Pt ring electrode by the CO? Have the authors performed CO stripping experiments on Pt ring electrodes?

3. The authors further compared the distribution of the gas phase products at different rotational speeds (different mass transfer conditions). When the mass transfer was enhanced, ethylene and ethane could be tested, while when the mass transfer was weakened, no relevant products were tested. However, why was the H₂ never detected? What is the mechanism of enhanced ethylene and ethane

production by accelerated mass transfer? The authors have not stated at which pH their mass spectrometry data were collected (ATR-SEIRAS and Raman test below are likewise not indicated). Did the authors use the RDE-GC-MS to compare the distribution of the products at different pHs and their corresponding onset potentials? How the authors stabilized the ionic strength of the electrolyte when regulating pH?

4. The authors carefully analyzed the Raman and IR spectra, especially the assignment of these peaks with Raman shifts located between 600 cm^{-1} and 1800 cm^{-1} . But why are we unable to observe a significant stark tuning? It is worth mentioning the Raman peaks located at 2800 cm^{-1} to 3000 cm^{-1} , which are frequently observed and may be exogenous species (DOI: 10.1021/acs.jpcc.7b03910). Did the authors perform a blank experiment?

5. In the Supporting Information, the authors give a schematic of the Raman electrolyzer, but this schematic has appeared in other literature and is almost identical (except for the erasure of some easily removed text, DOI: 10.1021/jacs.0c02354). However, the authors fail to make any citation or explanation. It is strongly recommended that the authors redraw the schematic.

6. There are some writing errors, for example, it should be Fig. 3 instead of Fig. 2 (Line 233, Page 12).

Reviewer #2 (Remarks to the Author):

In this work, RRDE-GC-MS, in conjunction with in situ vibrational spectroscopy and DFT calculations, is applied to study the reduction of vinyl alcohol. In general, this work is of novelty and interest, providing a new insight into the electrochemistry of enols in aqueous electrolyte. Yet, there remains a serious concern about the design of the experiment, in which the authors overlook the importance to stabilize the interfacial pH. For details, see the following comment.

1. Several important arguments by the authors are inferred from the pH effect and mass transport effect on the electrocatalytic reduction. However, the experiment conditions were poorly controlled, which makes these arguments less convincing. According to a recent report (Anal. Chem. 2021, 93, 1976–1983), even under the RDE configuration, the local pH at the electrode/electrolyte interface can dramatically deviate from the bulk pH in a non-buffered solution during a proton producing or consuming reaction, thus may lead to a misinterpretation of the experimental data. The variation of surface pH is even more severe when the RDE is not adopted, for instance, in the case of Raman and ATR-SEIRAS measurements. Unfortunately, in this work, (spectro)-electrochemical measurements were carried out in 0.1 M ClO_4^- + 0.1 M acetaldehyde solutions of pH 4, 7 and 10 without a buffering capacity. Therefore, the corresponding measurements should be re-conducted in pH-buffered solutions, unless they present very strong evidences and arguments against the major concern.

Additional concerns:

The conditions such as mass transport, electrode form and applied potential range are quite different in the RRDE and the spectroelectrochemical measurements, which makes the correlation of the two sets of data difficult. In the RRDE measurement, well-defined force convection was applied while in the ATR-SEIRAS and Raman measurements, no forced convection was applied; In the RRDE experiment, the bulk Cu was used as the working electrode while in ATR-SEIRAS and Raman measurements, the Cu powder was used as the WE; The onset reduction potential is -0.75 V vs RHE in the RRDE experiment while in ATR-SEIRAS and Raman spectroscopy, the potential range is -0.3 ~ 0.3 V.

Besides, isotope labeling is suggested in ATR-SEIRAS and Raman measurements for reliable band assignments.

Reviewer #3 (Remarks to the Author):

The electroreduction of acetaldehyde over a polycrystalline Cu electrode is examined. Rotating ring disk electrode (RRDE), in situ Raman and IR spectroscopy, and DFT calculations are included to probe reduction chemistry. Though acetaldehyde electroreduction on Cu has been previously studied, the new contribution to this work is indication that a different (not ethanol) reduction product(s) is formed at high overpotentials, indicated by a new oxidation signature in the Pt ring current. This product is speculated to be ethylene and/or ethane, which is also observed in GC-MS.

Though the overall direction of study is interesting, a few major concerns prevent recommendation of publication in the current form. These include both inherent limitations in the work and places where the presentation or basis for conclusions is unclear.

1) Experimentally, the assignment of the observed new product to ethylene/ethane is not conclusively demonstrated. An inherent limitation to the work is an inability to be quantitative in determining the distribution of products formed, including this "new" product. As such, it is unclear the extent to which the GC-MS observation is simply observing a small trace product, and its attribution to formation at a specific voltage, and the faradaic efficiency to this product, is unclear. Perhaps more importantly, the attribution of the low potential (~0.3 V-RHE) oxidation wave observed on the Pt ring to ethane/ethylene is quite suspect, and has little basis in the work. The indication that ethane and ethylene are oxidized on Pt in the 0.3-0.6 and 0.6-0.8 V ranges is attributed to a 1993 PhD thesis (which is challenging to find – I could not find a 1993 PhD thesis by Q. Zhao from CWRU, though the citation information is limited and unclear why the location is Ann Arbor). To attribute these oxidation waves to ethylene or ethane, the authors should cite more reliable (and accessible) evidence, and/or include new experimental data demonstrating this behavior.

2) The assignment of Raman and IR peaks to adsorbed vinyl alcohol intermediates is difficult to follow. Figure 3a does not provide a legend. It does not appear the legend to 3b applies to 3a, and the crossing of spectra in 3a makes observation of peaks (and figuring out what potential they occur at) challenging. Throughout Figures 3a-c, labeling of peaks would make connection of peak assignments to the text

easier to follow. Evidence for the adsorbed vinyl species, based on discussion in the text, appear to be a 3045 band in 3a, 1558 and 1520 bands in 3b, 1522 in 3c. None of these bands are easily distinguishable in the figures. Is the 3045 band observed in the ATR-SEIRAS – DFT binding geometries seem to suggest the CH₂ group is not entirely in the surface plane and should be somewhat IR active?

3) The DFT analysis does not add anything significant in reaching a mechanistic conclusion. Modeling electrochemical processes in DFT calculations is challenging due to the complexity of an electrochemical interface. Specific concerns relevant to this ability of this work to reach useful conclusions:

a. The adsorption energy of vinyl groups appears to be calculated in a manner that ignores the potential dependent interaction of water with the surface. As vinyl adsorption would actually be a replacement reaction, the minor change in adsorption energy with potential is not conclusively supported by the approach used to calculate the potential dependent adsorption energy. This might be improved by including potential dependent water displacement, but in practice, the use of DFT to calculate potential dependent adsorption of molecules that will surely interact with the solvent is very approximate, and, in this case, adds little useful in supporting conclusions in this work.

b. The reaction energetics of adsorbed vinyl alcohol reduction don't provide significant information in supporting the proposed reduction paths to ethylene and ethane. First, the energetics included for reduction of vinyl alcohol to ethanol are quite favorable, and there is no discussion as to why reduction to ethylene or ethane would be able to compete with the favorable and relatively straightforward paths to ethanol. Secondly, the key energetics in examining the path concluded by the authors to ethane and ethylene are not calculated and reported in this work. Page 14 sites prior work suggesting the barrier to direct hydrogenation of the hydroxyl group is high, however, calculation of barriers for coupled proton-electron transfer barriers is quite challenging in DFT calculations and simple citation to this work is far from sufficient to disallow this path. Solvation and double-layer effects can be large on such barriers, and critical evaluation of the possible paths is needed to disallow this step. The non-PCET path for reduction of adsorbed vinyl proposed in this work is missing calculation of its energetics, making conclusion that it is the dominant path so speculative as to make inclusion of the other DFT rather useless. First, even if the decoupled path would be operable, the sum of the reaction energies of the two decoupled steps would have to add up to the reaction energy of the coupled step, and even this sum of the two steps is not included. Were this to be unfavorable, then the decoupled path would have to be collectively uphill and likely inoperable. Were the coupled reaction energy favorable (and since no barriers are included for PCET steps in this work), then it is unclear why one would include the path is likely decoupled.

c. Following on the prior concerns, I note a separate concern as to whether an electron transfer to an adsorbate directly bound to the electrode surface can viably be considered as an elementary step to account for the speculated RDS in adsorbed vinyl reduction. The states to which electron transfer would occur (presumably a π^* state) are highly involved in bonding to the electrode, and likely form delocalized states residing at or near the fermi level of the coupled Cu-adsorbed vinyl system. As such, it is unclear to what extent a single potential would be crossed at which this state would suddenly become occupied, and more likely that pushing the Cu electrode negatively progressively increases the occupation of these states. Without DFT consideration of what is speculated as the distinguishing mechanistic feature of the work, the DFT work in this manuscript is collectively adding more confusion than useful addition to the study.

4) The discussion of the role of mass transfer and solution phase tautomerization to the enol in dictating the reduction paths is rather nebulous, and hard to allocate then to the mechanism proposed. It appears the authors are assuming all acetaldehyde/vinyl alcohol conversion would occur far from the surface, rather than the adsorbed vinyl alcohol species formation being part of a surface catalyzed reaction path. The basis for this assumption should be stated, as it is unclear why the opposite (ie, Cu promoting vinyl alcohol formation through surface reaction processes due to its favorable formation) could not be assumed. Further, the discussion of mass transfer effects appears to assume the tautomerization in solution is much slower than the transport of acetaldehyde or the enol to the surface, without providing explanation for this assumption.

Response to reviewer's comments:

We appreciate the reviewers for their constructive comments. These comments have helped us improve the quality of our manuscript significantly. In this revision, we have included additional experiments and explanation to address the reviewers' concerns and suggestions.

The original reviewer's comments are in **bold**. Our responses are in **blue**. The references are underlined.

Reviewer #1

The authors report the electrochemical conversion of vinyl alcohol and acetaldehyde on polycrystalline Cu to ethanol, ethylene and ethane; and propenol and propionaldehyde to propanol, propene and propane. Sensitive detection was achieved using a rotating disk electrode coupled with gas chromatography-mass spectrometry (RDE-GC-MS). In situ attenuated total reflection surface-enhanced infrared absorption spectroscopy (ATR-SEIRAS), and in situ Raman spectroscopy confirmed the adsorption of the vinyl alcohol. Calculations using canonical and grand-canonical density functional theory (DFT and GC-DFT), along with experimental findings suggest that the rate-determining step (RDS) for ethylene and ethane formation is an electron transfer step to the adsorbed vinyl alcohol. The experimental phenomena and the experimental conclusions are very interesting, and indeed innovative reaction mechanisms are proposed. Therefore, I recommend the publication in Nat. Commun. upon revisions

1.1. The authors used Pt ring electrodes for the identification of acetaldehyde reduction intermediates. The authors also mentioned that acetaldehyde in solution would adsorb to the Pt ring electrode surface and be oxidized to give CO₂ and acetic acid (Line 97, Page 4). It is possible that CO₂ may reabsorb on the surface of the Cu disc electrode and be reduced to obtain products such as ethylene. How did the authors avoid the reduction of CO₂ on the Cu surface during the test? Or how did the authors decouple the reduction current of CO₂ and the reduction current of acetaldehyde (enol) during the test?

We thank the reviewer for this comment. In our RRDE experiment, it is possible that oxidation products such as CO₂ can be generated on the Pt ring electrode in electrolytes containing 100 mM acetaldehyde and propionaldehyde (<https://doi.org/10.1021/la00092a006>). However, we do not believe that the ethylene and ethane detected are from the electroreduction of reabsorbed CO₂ on Cu disk and here are four reasons to support this response:

First, if ethylene and ethane were from the electroreduction of reabsorbed CO₂ on Cu, we should also see the same oxidation peaks for both the acetaldehyde and propionaldehyde oxidation, as both aldehydes should yield CO₂ at high oxidation potentials on Pt (<https://doi.org/10.1021/la00092a006>). However, our RRDE experiment for propionaldehyde (Fig. R1 or Supplementary Fig.7, also copied below) does not have the same ethylene/ethane oxidation feature. The oxidation features of propionaldehyde are confirmed by the RDE-GC-MS to be propane and propene with no ethane or ethylene

detected. Thus, it is very unlikely that ethylene and ethane detected was from the electroreduction of reabsorbed CO_2 on Cu.

Fig. R1 | RRDE measurements during propionaldehyde reduction on Cu disk electrode. Electrolyte: 0.1 M NaClO_4 containing 100 mM propionaldehyde. CV conditions: 0 to 1.6 V and then back to 0 V (vs RHE), 50 mV/s, 1600 RPM, 298 K. Only the positive-going scans were shown in CVs for clarity.

Second, all the cyclic voltammograms were collected while the solution is actively purged with nitrogen, so the amount of dissolved CO_2 resulting from a previous scan should be negligible. The likelihood of reabsorbing this CO_2 is very low. This deduction is supported by two experimental evidence. Our RDE-GC-MS experiments shows that ethylene and ethane are produced on Cu from the reduction of acetaldehyde, in the absence of an oxidation reaction (hydrocarbon to CO_2) occurring at a second electrode (i.e. no Pt ring on the RDE-GC-MS setup). Additionally, CO_2 reduction on a planar, pure Cu catalysts does not yield ethane. Therefore a reabsorbed CO_2 reduction will not yield ethane. (<https://doi.org/10.1039/F19898502309>; <https://doi.org/10.1039/C2EE21234J>).

Third, the ring data presented in this work is from the first CV scan. The reproducibility of the system was confirmed by repeating the experiments multiple times after repolishing the catalysts before each experiment. The onset potential for the oxidation of ethane and ethylene (> ca. 0.35 V vs. RHE at pH =7) on the ring electrode was much lower than the oxidation potential of acetaldehyde to CO_2 (> 0.9 V vs. RHE at pH=7). Which means, we can detect ethylene and ethane, before CO_2 is even produced from acetaldehyde oxidation.

Fourth, CO_2 reduction on Cu will give rise to many products besides ethylene (i.e. hydrogen, methane, carbon monoxide, formate, ethanol ...). If the electroreduction of reabsorbed CO_2 is occurring, the product distribution will show a very dominant CO peak at ca. 1 V vs. RHE on the Pt ring, as reported in our previous work (*J. Electrochem. Soc.* **167** 046517). In this work, CO oxidation feature is not observed on the Pt ring during the RRDE experiment. We therefore conclude that without the presence of a CO oxidation feature, it is very difficult to justify that CO_2 reduction is being reabsorbed and occurring on the Cu disk and that the ethylene formation is from the reduction of reabsorbed CO_2 .

Based on the above arguments, we do not think that the observed ethylene and ethane products are a result of the electroreduction of reabsorbed CO₂ on Cu. In the occasion that CO₂ is produced from the oxidation processes, the corresponding partial reduction current from reabsorbed CO₂ on the Cu catalysts will be much smaller compared to the acetaldehyde(enol) reduction current. The resulting CO₂ products will correspondingly be extremely small and negligible.

1.2. The authors mentioned that no oxidation current of ethanol was detected in the Pt ring (Supplementary Fig. 3). Such a phenomenon is interesting and critical. With such a phenomenon, the authors suggest that the observed oxidation peaks located at 0.3 V vs. RHE and 0.9 V vs. RHE should be species other than acetaldehyde and ethanol (i.e., ethylene and ethane). However, it is still difficult for me to understand why the oxidation of ethanol is not detected. Acetaldehyde is an intermediate for ethanol production (DOI: 10.1002/anie.201508851; 10.1021/jacs.8b04058), so the concentration of ethanol should be higher than other electrolysis products. In addition, there are various reactions for the oxidation of ethylene, which may give CO₂, ethylene oxide (10.1126/science.aaz8459), ethylene glycol (DOI: 10.1038/s41929-019-0386-4) and etc. Thus, which reaction does the oxidation peak detected by the authors correspond to? Meanwhile, the thermodynamic potential of the above mentioned oxidation reaction of ethylene is not always more negative than that of ethanol (e.g., 1.0 V vs. RHE for the oxidation of ethylene to ethylene oxide). If these oxidation reactions can be detected, why is the oxidation of ethanol not detected? Is there a poisoning of the smooth Pt ring electrode by the CO? Have the authors performed CO stripping experiments on Pt ring electrodes?

We thank the reviewer for raising this important point. First, we focus on why electrooxidation of ethanol was not detected. According to a previous study on the electrooxidation mechanism of acetaldehyde and ethanol on polycrystalline Pt electrodes reported by another group (<https://doi.org/10.1021/jp807350h>; [https://doi.org/10.1016/0013-4686\(90\)90032-U](https://doi.org/10.1016/0013-4686(90)90032-U)), both ethanol and acetaldehyde undergo dissociative adsorption at low overpotentials (ca. 0.1 V_{RHE} in acidic media) on Pt surface, which will result in adsorbed C1 species such as CO and CH_x. These species can be further oxidized to CO₂ at high overpotentials. However, the amount of the adsorbed C1 species is higher for acetaldehyde than for ethanol oxidation. Also, C-C scission is faster in acetaldehyde compared to ethanol. So, we can expect that dissociative adsorption of acetaldehyde is more favorable than that of ethanol on Pt. This point can be further supported by another published work (<https://doi.org/10.1016/j.jelechem.2006.10.009>), which revealed that the dissociative adsorption of ethanol is inhibited by adsorbed hydrogen on Pt. However, dissociated acetaldehyde can displace adsorbed hydrogen resulting in a relatively high surface coverage of hydrocarbon fragments. We believe that dissociative adsorption of acetaldehyde is both thermodynamically and kinetically favorable compared to ethanol dissociation on Pt. These results are consistent with our Pt ring CVs data (Supplementary Figure. 3) which shows that adding 10 mM ethanol to 100 mM acetaldehyde does not change the acetaldehyde oxidation current on Pt in both forward and backward scan. We believe that this is due to Pt surface being covered by the C1

species from the thermodynamically favorable acetaldehyde dissociation on Pt. Although ethanol is the primary electroreduction product from acetaldehyde, oxidation feature of ethanol on Pt was not detected at low ethanol concentration due to the competing faster and more favorable dissociative acetaldehyde adsorption, as well as the relatively high concentration of acetaldehyde used in this work compared to the ethanol produced.

Second, we provide literature evidence of ethylene oxidation on Pt in aqueous solution and how this relates to the voltametric features observed on our RRDE experiment. It was reported that Pt is a very good catalyst for oxidizing ethylene. A complete electrooxidation of ethylene to CO₂ was achieved in acidic media at 40-80 °C (<https://doi.org/10.1149/1.2426221>; <https://doi.org/10.1021/j100894a019>). As shown in Fig. R2 (a), (b) and (c), cyclic voltammograms of platinumized polycrystalline Pt with different ethylene saturation pressures were reported in a previous work (<https://doi.org/10.1149/1.2129765>). The onset potential of the electrooxidation of ethylene is ca. 0.4 V_{RHE} in acidic media, which is close to our observed onset potential of ca. 0.35 V_{RHE}. The adsorption of hydrogen is hindered by the surface adsorbed ethylene, which indicates that ethylene tends to bind to Pt strongly. Another way of putting it, adsorption of ethylene on Pt seems to be stronger than the adsorption of hydrogen and ethanol on Pt. In the same literature, it was also reported that the potential window for ethylene electrooxidation is quite broad, from ca. 0.4 V_{RHE} to 1.5 V_{RHE} on Pt. The position of the anodic oxidation peaks was also observed to shift as a function of ethylene partial pressure. Based on the reported literature, it is very challenging to assign specific oxidation reaction and adsorbed species to each of the ethylene oxidation peaks. However, we can assert that while a complete oxidation of ethylene will yield CO₂, a partial oxidation may yield some surface adsorbed and non-adsorbed organic intermediates as the reviewer pointed out in this comment.

Fig. R2 | CVs of Polycrystalline Pt in the presence of **ethylene** with a partial pressure of **(a) 1 atm (b) 0.1 atm (c) 0.01 atm**, electrolyte: 2 M H₂SO₄, CV conditions: 0.3 to 1.5 V and then back to 0.3V (vs RHE), 400 mV/s, 333 K, the lower profile is the blank CV taken from N₂ sat. solution, adapted from DOI: <https://doi.org/10.1149/1.2129765>; CVs of Polycrystalline Pt in the presence of **(d) ethane** sat. 2 M H₂SO₄, CV conditions: 0.3 to 1.5 V and then back to 0.3V (vs RHE), 400 mV/s, 353 K, the lower profile is the blank CV taken from N₂ sat. solution, adapted from DOI: <https://doi.org/10.1149/1.2127200>; (e) CV of Pt (100) in the presence of **(e) ethane** sat. 0.1 M HClO₄, CV conditions: 0.05 to 0.9 V and then back to 0.05V (vs RHE), 50 mV/s, room temperature, the lower profile with black line is the blank CV taken from Argon sat. solution, adapted from DOI: <https://doi.org/10.1016/j.jelechem.2021.115252>.

Third, similar to ethylene oxidation, ethane electrooxidation was also reported to occur over a broad potential window ranging from ca. 0.4 V_{RHE} to 1.5 V_{RHE} on Pt (<https://doi.org/10.1149/1.2127200>), as shown in Fig R2 (d). It is therefore likely that in the presence of ethane and ethylene, the oxidation current of these two hydrocarbons will overlap. Which means that the oxidation peak observed from our RRDE ring data can result from an overlapping ethane and ethylene oxidation. This is the reason why we attempted to verify the products in a mass spectrometer, which confirmed the presence of ethane and ethylene.

It is interesting to note that ethane oxidation is very sensitive to surface structures according to a more recent study on the electrooxidation of ethane at low index single crystal Pt surfaces in acidic media (<https://doi.org/10.1016/j.jelechem.2021.115252>), as shown in Fig R2 (e). This paper reports that Pt(100) is most effective for oxidizing ethane among the low index surfaces. These results also indicate that adsorption of ethane and its dissociated species will block the surface sites for hydrogen adsorption on Pt at low overpotential. Thus, we can detect the oxidation current of ethane for the same reason as we detect oxidation of ethylene on Pt. But it is not possible for us to deconvolute the ethane or ethylene partial oxidation current from the voltammogram observed on Pt.

As shown in Fig. R3, it is important to note that the deactivation of ring electrode was not observed in our experiments. According to previous studies on the electrooxidation of acetaldehyde on polycrystalline Pt electrode (<https://doi.org/10.1016/j.jelechem.2006.10.009>; <https://doi.org/10.1021/acs.jpcllett.0c02558>), all adsorbed C1 species resulted from the dissociative adsorption of acetaldehyde such as CH_x and CO can be further oxidized to CO_2 at high overpotential on Pt. We assume that adsorbed CO and CH_x are oxidatively stripped from the surface as we cycle to a high electrode potential of 1.6 V_{RHE} . Here, we did not perform CO stripping experiment on the Pt ring since CO poisoning effect was not observed on our Pt ring. In this work, a clean and blank Pt ring CV is regularly measured and can be easily reproduced after each experiment.

Fig. R3 | CV at Pt ring in the presence of acetaldehyde. Electrolyte: N₂ sat. 0.1 M NaClO₄ containing 25 mM acetaldehyde. CV conditions: 0 to 1.6 V and then back to 0 V (vs RHE), 100 mV/s, 1600 RPM, 298 K, 10 cycles.

Additional discussion on ethylene and ethane oxidation have been added to lines 170 to 181 (page 9 and 10) of the main text.

1.3. The authors further compared the distribution of the gas phase products at different rotational speeds (different mass transfer conditions). When the mass transfer was enhanced, ethylene and ethane could be tested, while when the mass transfer was weakened, no relevant products were tested. However, why was the H₂ never detected? What is the mechanism of enhanced ethylene and ethane production by accelerated mass transfer? The authors have not stated at which pH their mass spectrometry data were collected (ATR-SEIRAS and Raman test below are likewise not indicated). Did the authors use the RDE-GC-MS to compare the distribution of the products at different pHs and their corresponding onset potentials? How the authors stabilized the ionic strength of the electrolyte when regulating pH?

We thank the reviewer for this important comment. As we mentioned in the method section of the main text, our GC is equipped with two gas-sampling loops. The incoming sample is split in two and introduced to two gas sampling loops which leads to two different sets of separation and detection methods. Hydrocarbons were separated by a CP-PoraBond Q capillary column and separated molecules are detected by a quadrupole mass spectrometer, while hydrogen and CO are separated through a Haysep Q and Molsieve 5A column in series and the separated gasses are detected using a thermal conductivity detector (TCD). The experimental setup is described in detail in our previous publication (<https://doi.org/10.1021/acscatal.7b02373>).

We apologize for only showing the results for the hydrocarbons in the original manuscript. We have now included the H₂ data from the TCD detector in the revised manuscript, and shown in Fig. R4 below.

Fig. R4 | RDE-GC-MS measurements from thermal conductivity detector (TCD). Acetaldehyde reduction at -1.2 V vs RHE, (a) 1600 RPM; (b) 900 RPM; (c) no rotation; (d) Acetaldehyde reduction at -0.7 V vs RHE, 1600 RPM; (e) Acetone reduction at -1.2 V vs RHE, 1600 RPM; (f) Propionaldehyde reduction at -1.2 V vs RHE, 1600 RPM, in 0.1 M NaClO₄ solution saturated with N₂(pH=7).

Based on our TCD measurement, hydrogen was detected during acetaldehyde reduction at -1.2 V_{RHE} with a rotation rate of 1600 and 900 RPM. Hydrogen was also detected during acetone reduction at -1.2 V_{RHE} with a rotation rate of 1600 RPM. Hydrogen was not detected in any of the other experiments described. A detailed explanation for these results requires a comprehensive understanding on the competition reaction between small aldehydes/ketones reduction and hydrogen evolution reaction under a well-defined mass transport regime on polycrystalline Cu electrode. Our future work will focus on elucidating the mechanism of this competing reaction by systematically varying electrolyte parameters such as buffer strength, nature of anions and cations.

We ascribe the enhanced ethylene and ethane production at high overpotential to a faster mass transport by increasing the rotation rate of the electrode, and assuming that mass transport effect has negligible influence over the keto-enol equilibrium. That is, we assume that the equilibrium constant does not change as we change the rotation rate of the electrode. For an ideal mass transport limited condition in an RDE setup, the diffusion layer thickness at steady state can be estimated as follows (Bard, A. J., & Faulkner, L. R. (2000). *Electrochemical methods and applications*. New York: Wiley-Interscience):

$$\delta_F = 1.61 D_F^{\frac{1}{3}} \nu^{\frac{1}{6}} \omega^{-\frac{1}{2}}$$

where D_F is the diffusion coefficient of the reactant, ν is the kinematic viscosity of the solution and ω is the rotation rate. At a high rotation rate (≥ 1600 RPM), the diffusion layer thickness is around 1 μm which is 1-2 orders smaller than that obtained by varying the electrolyte flow rate or stirring the electrolyte solution (<https://doi.org/10.1021/acscatal.1c00272>; <https://doi.org/10.1021/jacs.9b10061>; <https://doi.org/10.1021/jacs.8b04058>). The diffusion layer thickness at 1600 rpm is even much smaller than the diffusion layer thickness under static condition (i.e. no rotation and no electrolyte flow). Here we argue that the precursor to ethylene and ethane is the adsorbed vinyl alcohol transported from the bulk solution. If true, one would expect that the extremely dilute vinyl alcohol will be very sensitive to the mass transport condition and its overall electrocatalytic reactivity (at high electrode potentials of -1.2 V_{RHE} shown in this work) will be limited by mass transport. As a result, we see more products (ethylene and ethane) formation by increasing the mass transport rate while these two products are not formed under static condition (no rotation) at the same electrode potential. After an extensive literature search, we also deem that the formation of ethylene and ethane from the electroreduction of acetaldehyde were not previously observed in the literature, most likely due to the experimental condition that does not typically involve high mass transport rates.

Our RDE-GC-MS, ATR-SEIRAS and Raman measurements were carried out in a neutral pH condition. The pH of all solutions was measured. The 0.1 M NaClO_4 electrolyte solution saturated with nitrogen or argon have a pH within the range of 6.8 to 7.0 before and after each measurement. In the revised manuscript, we have included details regarding the pH of the solution in the corresponding sections.

We used the RDE-GC-MS technique to identify the unknown products generated from the reduction of acetaldehyde (vinyl alcohol). However, we did not try to use RDE-GC-MS to compare the product distribution at different pH values or measure the onset potentials for

the formation of ethylene and ethane at each pH. As mentioned in the main text, line 157, page 9, our RDE cell coupled to GC-MS cannot be completely sealed and therefore quantification of products will not be appropriate. For example, gaseous products diffuse at different rates. They can leak through the gaps between the RDE cell wall and the rotating shaft. In addition to varying diffusion rates, the absence of (or a smaller) product signal in GC-MS result can be due to their electrocatalytic reactivity, or extent of leakage or both. As a result, we used RDE-GC-MS as a qualitative detection method to verify the compounds produced. The MS capability helps us determine the chemical identity of gaseous product, which works as a complementary technique to the more sensitive Pt ring detector on the RRDE.

As mentioned in the caption of Figure 1, page 8, in the RRDE experiment, the solution pH was regulated by adding the appropriate amount of acid or base to the supporting NaClO₄ solution while maintaining the same total ionic strength. For instance, the pH = 4 solution contains 0.0001M HClO₄+0.0999 M NaClO₄, the pH=7 solution contains 0.1 M NaClO₄, and the pH=10 solution contains 0.0001 M NaOH+0.0999 M NaClO₄. We have added additional description in the experimental section to improve the clarity of our experimental conditions.

Supplementary figure 14 has been added to the supplemental information (Page 10); the caption of supplementary figure 15 has been updated to include pH condition; additional description have been added to line 482,483,484 and 493 page 26; pH values have been added to the caption of Figure 2, page 12 and Figure 3, page 16; Additional description have been added to lines 89 and 90, page 4; additional discussions have been added to line 203 to 209.page 11 to make clarify the mechanism of ethylene and ethane production.

1.4. The authors carefully analyzed the Raman and IR spectra, especially the assignment of these peaks with Raman shifts located between 600 cm⁻¹ and 1800 cm⁻¹. But why are we unable to observe a significant stark tuning? It is worth mentioning the Raman peaks located at 2800 cm⁻¹ to 3000 cm⁻¹, which are frequently observed and may be exogenous species (DOI: 10.1021/acs.jpcc.7b03910). Did the authors perform a blank experiment?

We thank the reviewer for this important comment. Vibrational stark effect (VSE) has been reported for many stretching vibrations such as $\nu(\text{O-H})$, $\nu(\text{N-H})$, $\nu(\text{S-H})$, $\nu(\text{P-O})$, $\nu(\text{C=O})$, $\nu(\text{C}\equiv\text{O})$, $\nu(\text{C}\equiv\text{N})$ and $\nu(\text{azide})$ (<https://doi.org/10.1021/ar500464j>; <https://doi.org/10.1021/acs.jpcllett.6b01342>). To our knowledge, VSE is commonly observed for $\nu(\text{C=O})$ and $\nu(\text{C}\equiv\text{O})$, but not very prominent for $\nu(\text{C-H})$ or $\nu(\text{C=C})$.

We have assigned Raman and IR bands from the in-situ measurements to two key adsorbates on the Cu surface: ethoxy intermediate (CH₃CH₂O*) and vinyl alcohol (CH₂=CHOH*). These adsorbates contain C-H, C-O, C=C and O-H bonds, but not C=O. We anticipate that we should be able to see the Stark tuning effect for the $\nu(\text{C=O})$ band as a function of applied potential if acetaldehyde adsorbs on Cu

(<https://doi.org/10.1021/acs.jpcclett.0c02558>). However, we do not observe any significant shift in the vibrational bands (i.e. VSE) observed in our study. Also, it has been reported that acetaldehyde does not tend to adsorb on the Cu surface under typical electrocatalytic reduction conditions (<https://doi.org/10.1002/anie.201508851>). Therefore, it was not surprising that C=O is not observed.

Although vinyl alcohol contains OH group, the absence of O-H stretching vibration signal from vinyl alcohol in our IR and Raman measurement is due to overlap between the broad O-H stretching signal from H₂O and the O-H group from the alcohol.

The reviewer raises the question of exogenous species in the range of (2800-3000 cm⁻¹). First, we repeated the experiment by preparing the electrolyte solution in D₂O, as suggested by the reviewer in (DOI: 10.1021/acs.jpcc.7b03910). The D₂O data is shown in Fig. R16. Two bands appeared at 2183 cm⁻¹ and 2111 cm⁻¹ at -0.3 V which are assigned to the symmetrical C-D stretching (<http://dx.doi.org/10.1139/v56-178>; [https://doi.org/10.1016/05848539\(73\)80106-0](https://doi.org/10.1016/05848539(73)80106-0)) of CD₃ and CD₂ groups, respectively, from the adsorbed ethoxy intermediate. These groups may have been formed through surface H/D exchange as reported in previous study (<https://doi.org/10.1016/j.jcat.2018.11.019>). Collectively, the shifts in the Raman bands supports the assignment of 2800-3000 cm⁻¹ bands to adsorbed ethoxy, as opposed to the presence of an exogenous species. Moreover, if the bands at 2800 cm⁻¹ to 3000 cm⁻¹ were only related to exogenous species, we should see a decrease in signal in this region when D₂O is used as a solvent. Also, the fact that we observed C-D related bands at -0.3 V and no C-D related band at 0.05 V, suggests that vibrational bands in this region is due to the electroreduction reaction, and not from an exogenous species (as reported in DOI: 10.1021/acs.jpcc.7b03910).

Fig. R16 | Additional in situ Raman spectra obtained on polycrystalline copper powder electrode as a function of potentials in Argon saturated 0.1 M NaClO₄ (pH=7) with 100 mM acetaldehyde. The electrolyte is based on D₂O(left) and H₂O(right). All the potentials are

adjusted to the RHE scale.

Blank experiment was obtained in 0.1 M NaClO₄ as shown in Fig. R5, only perchlorate band is observed at 933 cm⁻¹. Again no exogenous species is detected in our 0.1 M NaClO₄ electrolyte solution. We include these blank spectra in our supporting document.

Fig. R5 | Blank in situ Raman spectra obtained on polycrystalline copper as a function of potentials in Argon saturated 0.1 M NaClO₄ (pH=7).

*Additional discussion have been added to Lines 207- 210, Page 11; 228-232, page 13.
Additional spectra have been added to supplementary figure 22 and 23.*

1.5. In the Supporting Information, the authors give a schematic of the Raman electrolyzer, but this schematic has appeared in other literature and is almost identical (except for the erasure of some easily removed text, DOI: 10.1021/jacs.0c02354). However, the authors fail to make any citation or explanation. It is strongly recommended that the authors redraw the schematic.

We highly appreciate this important advice and apologize for not including the citation to this figure. Our in-situ Raman flow cell was a commercial product purchased from Gauss Union, China. As the reviewer mentioned in the comment, this Raman cell was also used in a previous study (10.1021/jacs.0c02354), which has an almost identical schematic. We have redrawn our schematic of the Raman flow cell and included additional description and citation to the caption in Supplementary Fig 27.

Supplementary fig.27, page 19 has been updated.

1.6. There are some writing errors, for example, it should be Fig. 3 instead of Fig. 2 (Line 233, Page 12).

We thank the reviewer for this observation. We have updated Fig.2 to Fig.3.

Line 264 and 265 have been changed, page 14.

Reviewer #2

In this work, RRDE-GC-MS, in conjunction with in situ vibrational spectroscopy and DFT calculations, is applied to study the reduction of vinyl alcohol. In general, this work is of novelty and interest, providing a new insight into the electrochemistry of enols in aqueous electrolyte. Yet, there remains a serious concern about the design of the experiment, in which the authors overlook the importance to stabilize the interfacial pH. For details, see the following comment.

2.1. Several important arguments by the authors are inferred from the pH effect and mass transport effect on the electrocatalytic reduction. However, the experiment conditions were poorly controlled, which makes these arguments less convincing. According to a recent report (Anal. Chem. 2021, 93, 1976–1983), even under the RDE configuration, the local pH at the electrode/electrolyte interface can dramatically deviate from the bulk pH in a non-buffered solution during a proton producing or consuming reaction, thus may lead to a misinterpretation of the experimental data. The variation of surface pH is even more severe when the RDE is not adopted, for instance, in the case of Raman and ATR-SEIRAS measurements. Unfortunately, in this work, (spectro)-electrochemical measurements were carried out in 0.1 M ClO₄⁻ + 0.1 M acetaldehyde solutions of pH 4, 7 and 10 without a buffering capacity. Therefore, the corresponding measurements should be re-conducted in pH-buffered solutions, unless they present very strong evidences and arguments against the major concern.

We thank the reviewer for this comment, we agree with reviewer that local pH can deviate from bulk pH in our experiment. And, according to a previous work published by our group (<https://doi.org/10.1002/anie.201912637>), local pH can deviate dramatically from bulk pH even when using 0.1 M bicarbonate buffered solution during CO₂ reduction at high currents when the buffer capacity is reached, as shown in Fig. R6.

Fig. R6 | Local pH measured by IrO_x ring detector as a function of CO₂RR current measured on the Au disc. Red points: low CO₂RR current condition. Green points: high CO₂RR current condition, adapted from DOI: <https://doi.org/10.1002/anie.201912637>.

As we mentioned in the main text, although alkaline local pH may induce non-electrochemical reactions such as Cannizzaro disproportionation reaction in aqueous solution contained aldehydes (<https://doi.org/10.1021/jacs.6b12008>; <https://doi.org/10.1002/anie.201508851>), to the best of our knowledge, an increase in the local pH does not catalyze acetaldehyde to ethane or ethylene. To address the reviewer's concern, we repeated the RRDE, RDE-GC-MS, and Raman experiments in a pH 7, 0.1 M sodium phosphate buffer to verify and exclude the variable concerning drastic changes in the interfacial pH of an unbuffered solution, affecting the reaction intermediate and products from acetaldehyde reduction. RRDE experiment was conducted in a 0.1 M phosphate buffer solution (pH=7), as shown in Fig. R7. Figure 1, based on experiments conducted in 0.1 M NaClO₄ (pH=7) is copied from the main text for direct comparison. First, a similar oxidation feature is observed in 0.1 M phosphate buffer solution with an onset potential of ca. -0.8 V_{RHE}, which is very close to our measured onset potential (ca. -0.75 V_{RHE}) in 0.1 M NaClO₄ for the formation of ethane and ethylene. Here, we can conclude that the buffer capacity of the electrolyte solution does not influence our measured onset potential. We will include this buffered data to the supplemental information.

Fig. R7 | | Control experiment for RRDE. Right: Cyclic voltammograms (CVs) of products detected by the Pt ring electrode during Cu disk catalyzed acetaldehyde (100 mM) reduction at pH = 7 (0.1 M sodium phosphate buffer). Only the positive-going scans were shown for clarity. Cu powder potential was held from -0.4 V to -0.9 V. CV of the acetaldehyde blank was collected on the Pt ring when Cu disk was at open circuit potential (OCP). Left: Total Cu disk current during acetaldehyde reduction. CV conditions: 0 to 1.6 V and then back to 0 V (vs. RHE), 100 mV/s, 1600 RPM, 298 K.

Figure 1b and 1e in the main text. Data from 0.1 M NaClO₄ @ pH 7.

Figure 1. Rapid product detection by rotating ring-disk electrode (RRDE) during acetaldehyde reduction on Cu. a,b,c Cyclic voltammograms (CVs) of products detected by the Pt ring electrode during Cu catalyzed acetaldehyde reduction at pH = 4 (0.0001M HClO₄+0.0999 M NaClO₄), 7 (0.1 M NaClO₄), 10 (0.0001 M NaOH+0.0999 M NaClO₄), respectively. Only the positive-going scans were shown for clarity. Cu disk potential was held from -0.5 V to -1.2 V. CV of the acetaldehyde blank was collected on the Pt ring when Cu disk was at open circuit potential (OCP). d,e,f Total Cu disk current during acetaldehyde reduction. CV conditions: 0 to 1.6 V and then back to 0 V (vs. RHE), 100 mV/s, 1600 RPM, 298 K.

Second, the RDE-GC-MS experiments were reconducted in 0.1 M phosphate buffer with 100 mM acetaldehyde and relevant results are shown in Fig. R9, R10 and R11. In 0.1 M phosphate buffer solution, both ethylene and ethane can be detected at 1600 and 900 RPM when the disk was held at $-1.2 V_{RHE}$. Consistent with our 0.1 M $NaClO_4$ data, no ethylene or ethane could be detected at $-0.7 V_{RHE}$. These molecules were also not detected when the cell is not rotated even when a disc potential of $-1.2 V_{RHE}$ is applied.

Here we present that the buffered and unbuffered solution leads to the same conclusion we presented in our initial submission. We will again include data from the buffered solution experiments in the supplementary information so that this is available to the research community.

One main difference, however, is shown in Fig. R10, where hydrogen could be detected in all experiments conducted in a phosphate buffer solution, whereas no hydrogen is detected in the 0.1 M $NaClO_4$ (Fig. R4) at $-0.7 V_{RHE}$ or when no rotation was applied to the disk. This observation may be due to the buffering capacity of the phosphate buffer solution resulting in a lower surface pH, affecting the HER reaction.

Our results (in Fig. R8 and R9) are consistent with a recent report that a higher HER current and a less negative onset potential were observed in a buffer solution compared to the solution containing $NaClO_4$ without buffer (<https://doi.org/10.1016/j.jcat.2021.12.012>). However, in the presence of acetaldehyde, we agree that a more detailed explanation is needed to address the competing adsorption processes and electrocatalytic reaction between small aldehydes reduction and hydrogen evolution reaction under a well-defined mass transport regime. Our future work will focus on elucidating the mechanism of this competing reaction by investigating a matrix of variables and electrolyte parameters such as buffer strength, anions and cations.

Fig. R8 | | Linear sweep voltammetry (LSV) for acetaldehyde reduction on Cu. Electrolyte: 0.1 M NaClO₄ and sodium phosphate buffer (pH = 7) containing 100 mM acetaldehyde. LSV conditions: 0 to -1.0 V and then back to 0 V (vs RHE), 10 mV/s, 1600 RPM, 298 K. The current densities are normalized by electrode geometric areas.

Fig. R9 | Control experiment for hydrocarbons detection in buffer solution. Gaseous products generated from the electroreduction of acetaldehyde. Gas products were auto-injected via a gas sampling mechanism and analyzed using GC-MS in selected ion monitoring (SIM) mode at: **a** -1.2 V vs. RHE, 1600 RPM; **b** -1.2 V vs. RHE, 900 RPM; **c** -1.2 V vs. RHE, no rotation; **d** -0.7 V vs. RHE, 1600 RPM in 0.1 M phosphate buffer solution saturated with N₂(pH=7).

Fig. R10 | Control experiment for hydrogen detection in buffer solution. RDE-GC-MS measurements from thermal conductivity detector (TCD). Acetaldehyde reduction at -1.2 V vs RHE, (a) 1600 RPM; (b) 900 RPM; (c) no rotation; (d) Acetaldehyde reduction at -0.7 V vs RHE, 1600 RPM, 1600 RPM, in 0.1 M phosphate buffer solution saturated with N_2 (pH=7).

Fig. R11 | Control experiment for total current in buffer solution. Total current for the acetaldehyde reduction over Cu disk in 0.1 M phosphate buffer solution (pH = 7) at indicated conditions.

Third, we have remeasured the in-situ Raman in a 0.1 M phosphate buffer solution with 100 mM acetaldehyde, as shown in Fig. R12. It was reported (<https://doi.org/10.1021/jp970097k>; [https://doi.org/10.1016/S0022-0728\(01\)00542-3](https://doi.org/10.1016/S0022-0728(01)00542-3)) that adsorbed phosphates display several different Raman active vibrational modes in the frequency range from 400 cm^{-1} to 1200 cm^{-1} on Cu. We also observed these vibrations and highlighted them in yellow. Due to the adsorption bands of phosphate, reliable assignment of Raman signals will be more challenging in the lower frequency range in the phosphate buffer solution.

In this work, however, we can focus on the C-H stretching frequency range ($2700\text{--}3100\text{ cm}^{-1}$). In the phosphate buffer solution, vibrational band at 3046 cm^{-1} , 2920 cm^{-1} and 2860 cm^{-1} appeared at ca. 0 V_{RHE} , which is also observed at the potential values of $0.04\text{ V}_{\text{RHE}}$ and $0.07\text{ V}_{\text{RHE}}$ obtained in 0.1 M NaClO_4 solution. These bands were assigned to adsorbed vinyl alcohol and ethoxy intermediate, respectively. As a result, we can conclude that the same adsorbed intermediates are present in the phosphate buffer solution at a similar electrode potential on Cu. Although, it is not straightforward to assign the Raman bands located at a lower frequency range to any vibrational mode from adsorbed ethoxy or vinyl alcohol due to the specific adsorption of phosphate anions on Cu under our experimental condition. This set of data has also been added to supplemental information.

Fig. R12 | Control experiment for in-situ Raman experiment in buffer solution. These spectra were collected on polycrystalline copper powder electrode as a function of potentials in Argon saturated 0.1 M phosphate buffer solution (pH=7) with 100 mM acetaldehyde.

2.2 Additional concerns: The conditions such as mass transport, electrode form and applied potential range are quite different in the RRDE and the spectroelectrochemical measurements, which makes the correlation of the two sets of data difficult. In the RRDE measurement, well-defined forced convection was applied while in the ATR-SEIRAS and Raman measurements, no forced convection was applied; In the RRDE experiment, the bulk Cu was used as the working electrode while in ATR-SEIRAS and Raman measurements, the Cu powder was used as the WE;

We thank the reviewer for this comment, we agree with the reviewer that mass transport conditions are different in our RRDE and spectroscopy measurements. For an ideal mass transport limited condition at steady state, the diffusion layer thickness is around 1 μm at high rotation rates of (≥ 1600 RPM). This is about 1-2 orders of magnitude smaller than those obtained by varying the electrolyte flow rate or stirring the electrolyte solution. (<https://doi.org/10.1021/acscatal.1c00272>; <https://doi.org/10.1021/jacs.9b10061;10.1021/jacs.8b04058>).

In this work, 15 mL/min flow rate was used for the ATR-SEIRAS and Raman measurements. While the diffusion layer thickness of a 15 mL/min flow is different from a 1600 rpm rate, the main goal of the spectroscopy study is to figure out the potential-dependent adsorbates during the electroreduction of acetaldehyde on Cu. In our work, we observe the adsorption of vinyl alcohol under a much slower mass transfer condition. We foresee that this adsorption band could be much more pronounced if we were able to flow the solution at a much higher rate, which is not possible in the current spectroscopy setup and design. Here, we believe that the mass transport difference in the RRDE and spectroscopy measurements

does not alter our main conclusion that is the supporting evidence of the presence of vinyl alcohol, and therefore we propose that vinyl alcohol is the relevant precursor to ethylene and ethane products measured in this work. Though, it would be interesting to design a flow cell capable of higher flow rates or redesign a cell holder that is capable of integrating an RRDE to a spectrometer. This effort is however, beyond the scope of this study.

The reviewer also asks to discuss a comparison between Cu disk and Cu powders. Control experiments were carried out. Similar RRDE experiments were repeated with Cu powder and compared to the Cu disk in a 0.1 M NaClO₄ containing 100 mM acetaldehyde electrolyte solution. Data is shown in Fig. R13 where the Cu disk insert was replaced by a glassy carbon supported Cu powder (~6 mg) working electrode (WE).

The results from the Cu powder gave a similar onset potential of -0.7 V_{RHE} for the oxidation of ethylene and ethane on Pt. This result supports that surface morphology difference between planar Cu WE and Cu powder WE has little influence over the onset potential of ethane and ethylene. With this supporting data, we can conclusively state that our spectroscopy results from the Cu powder is directly comparable to the electrochemical results. However, according to our experience, one must be more careful when using Cu powder in a RDE setup, especially those containing high concentrations of acetaldehyde (i.e. 100 mM), since Cu powders can slowly delaminate from the glassy carbon substrate at the high rotation rate (≥ 1600 RPM), under high reducing potential in a solution containing significant amounts of organic compounds that can dissolve the binder. This limits the use of Cu powder electrodes for long-duration electrocatalysis study in this work. Figure R13 below has been added to the supplemental information.

Fig. R13 | Control experiment by using Cu powder as working electrode. Right: Cyclic voltammograms (CVs) of products detected by the Pt ring electrode during Cu powder catalyzed acetaldehyde (100 mM) reduction at pH = 7 (0.1 M NaClO₄). Only the positive-going scans were shown for clarity. Cu powder potential was held from -0.5 V to -1.0 V. CV of the acetaldehyde blank was collected on the Pt ring when Cu powder was at open circuit potential (OCP). Left: Total Cu powder current during acetaldehyde reduction. CV conditions: 0 to 1.6 V and then back to 0 V (vs. RHE), 100 mV/s, 1600 RPM, 298 K.

2.3 The onset reduction potential is -0.75 V vs. RHE in the RRDE experiment while in ATR-SEIRAS and Raman spectroscopy, the potential range is -0.3 ~ 0.3 V.

We thank the reviewer for this comment. We have collected both Raman and ATR-SEIRAS spectra between 0.3 to -0.8 V_{RHE}. We only showed -0.3 to 0.3 V in the initial submission since the spectra from -0.3 to -0.9 V are essentially similar. We have now added these spectra (Fig. R14 and Fig. R15 below) to the supporting information.

Fig. R14 | Additional in situ Raman spectra obtained on polycrystalline copper powder electrode as a function of potentials in Argon saturated 0.1 M NaClO₄ (pH=7) with 100 mM acetaldehyde. All the potentials are on the RHE scale.

Fig. R15 | Additional in situ ATR-SEIRAS spectra obtained on polycrystalline copper powder electrode as a function of potentials in Argon saturated 0.1 M NaClO₄ (pH=7) with 100 mM acetaldehyde. All the potentials are on the RHE scale.

These additional spectra are now in Supplementary Fig. 20 and 21.

2.4 Besides, isotope labeling is suggested in ATR-SEIRAS and Raman measurements for reliable band assignments.

We thank the reviewer for this comment. We agree with reviewer that isotope labeling will make band assignment more reliable. In this work, we conducted a similar Raman experiment in D₂O based electrolyte to confirm the band assignment of adsorbed ethoxy (10.1021/acs.jpcc.7b03910) in the range between 2800 cm⁻¹ and 3000 cm⁻¹.

The D₂O data is shown in Fig. R16. Two bands appeared at 2183 cm⁻¹ and 2111 cm⁻¹ at -0.3 V which are assigned to the symmetrical C-D stretching (<http://dx.doi.org/10.1139/v56-178>; [https://doi.org/10.1016/05848539\(73\)80106-0](https://doi.org/10.1016/05848539(73)80106-0)) of CD₃ and CD₂ groups, respectively, from adsorbed ethoxy intermediate. These groups may have been formed via a surface H/D exchange as reported in a previous study (<https://doi.org/10.1016/j.jcat.2018.11.019>). Collectively, the shifts in the Raman bands supports the assignment of 2800-3000 cm⁻¹ bands to adsorbed ethoxy, as opposed to the presence of an exogenous species. Moreover, if the bands at 2800 cm⁻¹ to 3000 cm⁻¹ were only related to exogenous species, we should see a decrease in signal in this region when D₂O is used as a solvent. The fact that we observed C-D related bands at -0.3 V and no C-D related band at 0.05 V, suggests that vibrational bands in this region is due to an electroreduction reaction, and not from an exogenous species (as reported in DOI: 10.1021/acs.jpcc.7b03910).

Fig. R16 | Additional in situ Raman spectra obtained on polycrystalline copper powder electrode as a function of potentials in Argon saturated 0.1 M NaClO₄ (pH=7) with 100 mM acetaldehyde. The electrolyte is based on D₂O(left) and H₂O(right). All the potentials are on the RHE scale.

Additional control experiments have been added to supplementary figures 2,4,5,13,16,17,19,20,21,22,23 and 24.

Additional descriptions have been added to Page 5, Line 118-121, which is also copied below:

It's important to note that a similar onset potential of ca. -0.8 V vs. RHE was also observed by conducting the same measurement in the 0.1 M phosphate buffer solution (pH = 7) as shown in Supplementary Fig.4.

Reviewer #3

The electroreduction of acetaldehyde over a polycrystalline Cu electrode is examined. Rotating ring disk electrode (RRDE), in situ Raman and IR spectroscopy, and DFT calculations are included to probe reduction chemistry. Though acetaldehyde electroreduction on Cu has been previously studied, the new contribution to this work is indication that a different (not ethanol) reduction product(s) is formed at high overpotentials, indicated by a new oxidation signature in the Pt ring current. This product is speculated to be ethylene and/or ethane, which is also observed in GC-MS.

Though the overall direction of study is interesting, a few major concerns prevent recommendation of publication in the current form. These include both inherent limitations in the work and places where the presentation or basis for conclusions is unclear.

3.1) Experimentally, the assignment of the observed new product to ethylene/ethane is not conclusively demonstrated. An inherent limitation to the work is an inability to be quantitative in determining the distribution of products formed, including this “new” product. As such, it is unclear the extent to which the GC-MS observation is simply observing a small trace product, and its attribution to formation at a specific voltage, and the faradaic efficiency to this product, is unclear. Perhaps more importantly, the attribution of the low potential (~0.3 V-RHE) oxidation wave observed on the Pt ring to ethane/ethylene is quite suspect, and has little basis in the work. The indication that ethane and ethylene are oxidized on Pt in the 0.3-0.6 and 0.6-0.8 V ranges is attributed to a 1993 PhD thesis (which is challenging to find – I could not find a 1993 PhD thesis by Q. Zhao from CWRU, though the citation information is limited and unclear why the location is Ann Arbor). To attribute these oxidation waves to ethylene or ethane, the authors should cite more reliable (and accessible) evidence, and/or include new experimental data demonstrating this behavior.

We agree with the reviewer regarding the inherent limitation of the current RDE-GC-MS

setup for quantifying the products and providing Faradaic efficiency (FE). We have another well-sealed electrochemical reactor that can provide quantitative product, however, this reactor is not able to provide fast enough flow rates mimicking those experienced at 1600 rpm in an RDE. Therefore, in this work, we have coupled our RDE to a GCMS (coined RDE-GC-MS) to be able to have the ability to conduct catalysis experiments at 1600 rpm. This setup, however, is qualitative due to difficulties in completely sealing the gap between the glass cell wall and the rotating shaft. Since gaseous species has varied diffusion rates, we are not comfortable quantifying the products from this setup to avoid further confusion. The purpose of the RDE-GC-MS is to confirm the presence and identity of ethylene and ethane at high mass transport rates. The detected amount of ethylene and/or ethane in our work represents a lower bound of the actual number of products produced.

At the time of our initial submission, we were only able to find reference to ethylene oxidation from a PhD thesis, although the thesis was very well written, and the conclusions are very well supported. We have since able to find additional literature report on ethane and ethylene oxidation. It was reported that Pt is a very good catalyst for oxidizing ethylene. A complete electrooxidation of ethylene to CO₂ can be achieved in acidic media at 40-80 °C (<https://doi.org/10.1149/1.2426221>; <https://doi.org/10.1021/j100894a019>). As shown in Fig. R2 (a), (b) and (c), cyclic voltammograms of platinized polycrystalline Pt with different ethylene saturation pressures were reported in a previous study (<https://doi.org/10.1149/1.2129765>). The onset potential of the electrooxidation of ethylene is ca. 0.4 V_{RHE} in acidic media, which is close to our observed onset potential of ca. 0.35 V_{RHE}. Moreover, in this paper, the authors report that the adsorption of hydrogen is hindered by surface adsorbed ethylene. This suggests that ethylene tends to bind to Pt surface strongly, and that ethylene adsorption is stronger than hydrogen adsorption on Pt. In the same manuscript, it was also reported that the potential window of the observed ethylene electrooxidation was quite broad, from ca. 0.4 V_{RHE} to 1.5 V_{RHE} on Pt, and that the position of anodic oxidation peaks will also shift as a function of ethylene partial pressure.

Fig. R2 | CVs of Polycrystalline Pt in the presence of **ethylene** with a partial pressure of **(a) 1 atm (b) 0.1 atm (c) 0.01 atm**, electrolyte: 2 M H₂SO₄, CV conditions: 0.3 to 1.5 V and then back to 0.3V (vs RHE), 400 mV/s, 333 K, the lower profile is the blank CV taken from N₂ sat. solution, adapted from DOI: <https://doi.org/10.1149/1.2129765>; CVs of Polycrystalline Pt in the presence of **(d) ethane** sat. 2 M H₂SO₄, CV conditions: 0.3 to 1.5 V and then back to 0.3V (vs RHE), 400 mV/s, 353 K, the lower profile is the blank CV taken from N₂ sat. solution, adapted from DOI: <https://doi.org/10.1149/1.2127200>; (e) CV of Pt (100) in the presence of **(e) ethane** sat. 0.1 M HClO₄, CV conditions: 0.05 to 0.9 V and then back to 0.05V (vs RHE), 50 mV/s, room temperature, the lower profile with black line is the blank CV taken from Argon sat. solution, adapted from DOI: <https://doi.org/10.1016/j.jelechem.2021.115252>.

Similar to ethylene oxidation, ethane electrooxidation was also reported to occur over a broad potential window range of 0.4 V_{RHE} to 1.5 V_{RHE} on Pt (<https://doi.org/10.1149/1.2127200>), as shown in Fig R2(d). The oxidation of ethylene and ethane, however, occurs over the same potential window. Therefore, in our work, the oxidation current of ethylene and ethane cannot be easily deconvoluted. It is interesting to note that ethane oxidation is very sensitive to surface structures according to a more recent

study on the electrooxidation of ethane on different low index single crystal Pt surfaces in acidic media (<https://doi.org/10.1016/j.jelechem.2021.115252>), as shown in Fig R2(e). The authors report that Pt(100) is most effective for oxidizing ethane among low index surfaces. The study also indicate that adsorption of ethane and its dissociated species block the surface site for hydrogen adsorption at the low potential region.

It is also interesting to note that propene oxidation also occurs over a broad potential window ranging from 0.6 to 1.6 V_{RHE} on polycrystalline Pt in acidic media (<https://doi.org/10.1021/jp021726f>) **Fig. R17**. This oxidation region also corresponds to the observed onset oxidation potential (0.6 V_{RHE}) of propene/propane from the propionaldehyde reduction product using the same RRDE setup.

Fig. R17 | (a) RRDE measurements during propionaldehyde reduction on Cu disk electrode. Electrolyte: 0.1 M NaClO_4 containing 100 mM propionaldehyde. CV conditions: 0 to 1.6 V and then back to 0 V (vs RHE), 50 mV/s, 1600 RPM, 298 K. Only the positive-going scans were shown in CVs for clarity. (b) CV of Polycrystalline Pt in the presence of propene with a concentration of 20 mM, electrolyte: 0.25 M H_2SO_4 , CV conditions: 12.5 mV/s, room temperature, the x-labeled profile is the blank CV taken from Argon sat. solution, adapted from DOI: <https://doi.org/10.1021/jp021726f>.

Additional discussions have been added to the main manuscript to improve clarity (line 164-175, page 8-9)

3.2) The assignment of Raman and IR peaks to adsorbed vinyl alcohol intermediates is difficult to follow. Figure 3a does not provide a legend. It does not appear the legend to 3b applies to 3a, and the crossing of spectra in 3a makes observation of peaks (and figuring out what potential they occur at) challenging. Throughout Figures 3a-c, labeling of peaks would make connection of peak assignments to the text easier to follow. Evidence for the adsorbed vinyl species, based on discussion in the text, appear to be a 3045 band in 3a, 1558 and 1520 bands in 3b, 1522 in 3c. None of these bands are easily distinguishable in the figures. Is the 3045 band observed in the ATR-SEIRAS – DFT binding geometries seem to suggest the CH_2 group is not entirely in the surface plane and should be somewhat IR

active?

We thank the reviewer for this helpful comment, we have replotted Figure 3 to improve clarity, as shown below as Fig. R18

Fig. R18 | (a) In situ Raman and (b) ATR-SEIRAS spectra (background spectrum was recorded at 0.4 V) obtained on polycrystalline copper as a function of potentials in Argon saturated 0.1 M NaClO₄ (pH=7) with 100 mM acetaldehyde. (c) GC-DFT calculated potential dependent adsorption free energy of vinyl alcohol on Cu(100) and Cu(211), respectively.

We agree with the reviewer that $\nu(\text{=C-H})$ band may also be active in the IR. However, compared to Raman spectroscopy, IR spectroscopy is very responsive to bulk water. The IR-active $\nu(\text{=C-H})$ vibrational band may overlap with the broad $\nu(\text{O-H})$ band from water as shown in Fig. R19, which limits the use of ATR-SEIRAS for identifying species around the high vibrational frequency region in aqueous media. Moreover, the lower bound of our spectral

window in ATR-SEIRAS is limited to $\sim 1300\text{ cm}^{-1}$ due to a strong IR absorption of the Si ATR crystal in the lower vibrational frequency region (<https://doi.org/10.1021/acs.jpcc.8b09598>; <https://doi.org/10.1038/s41467-022-30262-2>).

Fig. R19 | ATR-SEIRAS spectra in high wavenumber range (2600 cm^{-1} to 3500 cm^{-1}) at indicated conditions.

3.3) The DFT analysis does not add anything significant in reaching a mechanistic conclusion. Modeling electrochemical processes in DFT calculations is challenging due to the complexity of an electrochemical interface. Specific concerns relevant to this ability of this work to reach useful conclusions: a. The adsorption energy of vinyl groups appears to be calculated in a manner that ignores the potential dependent interaction of water with the surface. As vinyl adsorption would actually be a replacement reaction, the minor change in adsorption energy with potential is not conclusively supported by the approach used to calculate the potential dependent adsorption energy. This might be improved by including potential dependent water displacement, but in practice, the use of DFT to calculate potential dependent adsorption of molecules that will surely interact with the solvent is very approximate, and, in this case, adds little useful in supporting conclusions in this work.

We thank the reviewer for this important comment. We agree with the reviewer that a more accurate computational model would be needed for taking explicit water molecules and ions into consideration, to model the possible water replacement reaction near the electrochemical interfacial region at constant electrode potential.

However, maintaining a constant electrode potential while modeling surface water replacement reaction at the same time is not trivial. Our study, therefore, does not attempt to provide an accurate estimation of adsorption energetics of vinyl alcohol at each electrode potential. All we seek is to compare the relative trends to support the spectroscopic observations. As the reviewer mentioned, we ignored the potential dependent interaction of water with the Cu surface in this work. Instead, we only focus on surface charge effect because in our experiments, the electrode surface is negatively charged at the cathodic potential. In this work, we use a simplified model which treats solvent effect as a potential independent external correction (<https://doi.org/10.1002/anie.201301470>). Our goal is to try to build a qualitative relation between the adsorption of vinyl alcohol and surface electronic charge as the electrode potential is stepped to more negative values, and correlate this to spectroscopic observations. Experimentally, we observe an enhanced adsorption of vinyl alcohol on Cu surface as the potential is stepped to more negative values in both Raman and IR spectra, which can be partly due to the accumulation of surface electronic charge as the potential becomes more negative. This, we think, will result in a stronger dipole-charge interaction between the adsorbate and the Cu surface, thus enhancing the adsorption observed.

Overall, the experimental observations are consistent with our calculated adsorption trend for vinyl alcohol. The calculated trend also shows that the specific adsorption becomes more thermodynamically favorable on both Cu(100) and Cu(211) due to the accumulation of surface electronic charge. However, we acknowledge that a more accurate model is needed in future studies.

Additional discussions included in the main manuscript for clarity (line257-line263, page 14).

b. The reaction energetics of adsorbed vinyl alcohol reduction don't provide significant information in supporting the proposed reduction paths to ethylene and ethane. First, the energetics included for reduction of vinyl alcohol to ethanol are quite favorable, and there is no discussion as to why reduction to ethylene or ethane would be able to compete with the favorable and relatively straightforward paths to ethanol.

Secondly, the key energetics in examining the path concluded by the authors to ethane and ethylene are not calculated and reported in this work.

Page 14 sites prior work suggesting the barrier to direct hydrogenation of the hydroxyl group is high, however, calculation of barriers for coupled proton-electron transfer barriers is quite challenging in DFT calculations and simple citation to this work is far from sufficient to disallow this path.

Solvation and double-layer effects can be large on such barriers, and critical evaluation of the possible paths is needed to disallow this step. The non-PCET path for reduction of adsorbed vinyl proposed in this work is missing calculation of its energetics, making conclusion that it is the dominant path so speculative as to make inclusion of the other DFT rather useless. First, even if the decoupled path would be operable, the sum of the reaction energies of the two decoupled steps would have to add up to the reaction energy

of the coupled step, and even this sum of the two steps is not included. Were this to be unfavorable, then the decoupled path would have to be collectively uphill and likely inoperable. Were the coupled reaction energy favorable (and since no barriers are included for PCET steps in this work), then it is unclear why one would include the path is likely decoupled.

We thank the reviewer for their comment. First, we agree with the reviewer that our calculated energetics for vinyl alcohol reduction to ethanol are thermodynamically favorable, and we do produce ethanol from acetaldehyde reduction in our experiments. At low overpotential, the sole reduction product from acetaldehyde reduction is ethanol. Our proposed competing reaction mechanism is to rationalize the experimentally observed ethylene/ethane products from acetaldehyde at high overpotentials and high mass transport rates. The computational work is performed to support that a mechanism to ethylene/ethane is plausible.

Computational studies have shown that the ethane and ethylene pathway is likely to occur via the enol (vinyl alcohol) form, which is at equilibrium with acetaldehyde in the solution, as opposed to a direct acetaldehyde reduction. We further suspect that this enol to ethane or ethylene pathway is also possible during actual CO₂ reduction on copper-based catalysts, as acetaldehyde (and its enol tautomer) is a known intermediate observed on Cu during CO₂ reduction. We have included additional descriptions to the original manuscript (line 397) to clarify this concept. Again, without significantly high mass transport rate, only ethanol is detected from acetaldehyde reduction.

We acknowledge that accurate estimation of coupled proton-electron transfer barrier is quite challenging in DFT calculations. Thus, we try to provide more relevant experimental evidence, along with previous literature report to support the computational effort. For instance, our experimentally measured onset potential for the production of ethylene and ethane are independent of pH. This finding suggests that the rate determining step (RDS) may not involve proton transfer. If this is true, then, the C-O bond of vinyl alcohol must be first cleaved so that ethylene and ethane can be formed in the subsequent reduction steps. Thus, we assume that C-O bond scission is the RDS. This key step can be achieved through (1) a thermochemical step, (2) a CPET step via water as proton donor, or (3) a decoupled electron transfer step.

First, we rule out the possibility of a thermochemical step, which was reported in a previous study where C-O bond cleavage can occur via a thermochemical step on Pt electrode (<https://doi.org/10.1021/ja406655q;10.1038/s41929-019-0229-3>). However, our calculated NEB barriers (1.30 eV for Cu(100) and 1.14 eV for Cu(211)) suggest that a thermochemical C-O bond scission is kinetically hindered on Cu surfaces as mentioned in the main text. Our calculated trend also shows that a thermochemical C-O bond scission of vinyl alcohol is more favorable on Pt(100) terrace site (0.99 eV). However, when we conducted acetaldehyde reduction on a polycrystalline Pt disk at 1600 RPM, no ethylene or ethane is detected (Supplementary Fig.4), which suggest that a thermochemical C-O bond scission of vinyl alcohol is not feasible on Pt at room temperature. This leads us to believe that a thermochemical C-O bond scission is even more unfavorable on Cu under current experimental conditions.

Furthermore, according to reported literature ([10.1039/c1sc00277e](https://doi.org/10.1039/c1sc00277e); [10.1039/c2ee21234j](https://doi.org/10.1039/c2ee21234j); <https://doi.org/10.1021/jacs.7b06765>; [10.1039/c1sc00277e](https://doi.org/10.1039/c1sc00277e)), several compounds along the CO₂ reduction pathways, such as glyoxal (OHC-CHO), glycolaldehyde (OHC-CH₂OH), and ethylene glycol (CH₂OH-CH₂OH), which contains C-O bond are not precursors to the formation of ethylene (or ethane) in CO₂ reduction on Cu, which may suggest that thermochemical C-O bond scission to form ethylene (or ethane) is not favorable on these oxalates on Cu. All of the experimental evidence from the literature supports that thermochemical C-O bond scission is not favorable on Cu electrode under room temperature in aqueous media.

Second, we try to exclude the possibility of cleaving C-O bond through a CPET step. In this process, a key step is to add a proton coupled electron pair to the hydroxyl group of vinyl alcohol to cleave the C-O bond on Cu. If this is the rate limiting step, the reaction will be dependent on the H⁺ concentration, which is not observed experimentally.

A CPET step involving water as proton donor can also be excluded. We acknowledge that some literature report indicates water mediated reaction may be pH independent (<https://doi.org/10.1021/ja067950q>). This is, however, inconsistent with the reported experiments on water reduction (HER) in neutral and alkaline media on Cu, Au and Pt ([10.1016/0013-4686\(92\)87118-J](https://doi.org/10.1016/0013-4686(92)87118-J); [10.1149/1.2428530](https://doi.org/10.1149/1.2428530); <https://doi.org/10.1021/jacs.9b10061>; [10.1002/celc.201701316](https://doi.org/10.1002/celc.201701316); [http://dx.doi.org/10.1002/anie.202102803](https://doi.org/10.1002/anie.202102803); <https://doi.org/10.1016/j.jcat.2021.12.012>). If water reduction was indeed pH-dependent, then we could expect that a CPET step involving water as proton donor should also be pH-dependent.

The RRDE experiments carried out in this work ranges from acidic (pH = 4) to basic (pH=10) conditions with a well-defined high mass transport condition (RPM=1600). All of the measured onset potential for ethylene/ethane formation are pH independent. It is therefore inconsistent to consider water as proton donor to be involved in the rate limiting step. These experimental results agree with the computational study (<https://doi.org/10.1021/acscatal.7b03477>) we cited in the main text, which reported that direct hydrogenation to the OH group of vinyl alcohol is kinetically blocked on Cu(100). In summary, we exclude the possibility of C-O bond scission induced by a CPET step based on our experimental and computational evidence.

Third, the remaining plausible rate limiting step is a decoupled proton-electron transfer to the adsorbed vinyl alcohol. Based on the reviewer's suggestion, we have added the sum of the reaction energies of the two decoupled steps to our proposed reaction pathway. This step is denoted by the dashed arrow. The updated proposed pathway is copied below (Fig. R20):

Fig. R20 | Free energies of the initial (IS), transition (TS), and final states (FS) for the C-O bond-breaking processes of adsorbed vinyl alcohol on Cu(100), Cu(211), Pt(100), respectively. E_{act} values correspond to the calculated NEB barriers with ZPE correction. **d** Proposed reaction pathway for vinyl alcohol reduction to ethylene, ethane, and ethanol on Cu(100) and Cu(211). All ΔG values are given at 0 V vs. RHE based on CHE model. Solid arrows indicate proposed pathway and dashed arrow denotes unlikely step.

As shown in step 5 in Fig. R20, the reaction free energy sum of these two decoupled steps are 0.11 eV and 0.03 eV on Cu (100) and Cu(211), respectively. Our result agrees very well with a recent explicit solvent model based computational study on Cu(100) which reported the same reaction free energy change (step 5) of 0.10 eV on Cu(100) (<https://doi.org/10.1021/acscatal.1c01486>). These steps are considered to be kinetically blocked on Cu as we discussed before.

For the decoupled electron-proton transfer argument, we have to consider our experimentally obtained onset potential values, since computing the energetics of a negatively charged adsorbed intermediate on the electrode surface is quite complex and beyond the scope of our work.

Our experimentally measured onset potential of ca. -1.1 V vs. SHE can be regarded as an approximation of equilibrium potential for the decoupled electron transfer step, so that the reaction free energy for the decoupled electron transfer step can be expressed as $\Delta G_{\text{exp}} = 1.1 - 0.059 \cdot pH$ by assuming a typical Nernstian shift on the RHE scale at 298 K (we have also added this detail to the reaction pathway in Figure R20). Here we assume that this step is limited by thermodynamics and not kinetics. Because decoupled electron-proton transfer processes are often considered to be limited by thermodynamics, not kinetics in heterogeneous electrocatalysis as described in previous literature

<https://doi.org/10.1039/C3SC50205H>).

It is important to note that assuming a decoupled electron transfer is consistent with all our RRDE experimental results. For instance, at low reduction potentials, no ethylene or ethane is observed because the thermodynamic barrier for the first electron transfer step towards ethylene/ethane formation has not yet been overcome. At low overpotentials, the dominant pathway is ethanol formation. Once the reducing potential is negative enough to shift the electrochemical potential of the electrons to overcome the thermodynamic barrier of the electron transfer step towards ethylene, C-O bond can be cleaved by a decoupled proton-electron transfer process as shown in our reaction pathway. Under a high reducing potential used in this work ($-1.2 V_{\text{RHE}}$), all the free energy changes will be highly exergonic. Thus, we assume that the RDS of the reduction of vinyl alcohol to ethylene and ethane is a decoupled electron transfer step before the thermodynamic barrier is overcome.

To sum up, we acknowledge that a more precise and accurate consideration should also include potential dependent activation barrier, pre-exponential factor and explicit solvent effect for each elementary step, however, to the best of our knowledge, decoupled electron transfer is the most plausible mechanism that is consistent with our experimental results.

We have added these additional discussions to page 18, 19 and 20 to make our statement more convincing.

c. Following on the prior concerns, I note a separate concern as to whether an electron transfer to an adsorbate directly bound to the electrode surface can viably be considered as an elementary step to account for the speculated RDS in adsorbed vinyl reduction. The states to which electron transfer would occur (presumably a π^* state) are highly involved in bonding to the electrode, and likely form delocalized states residing at or near the Fermi level of the coupled Cu-adsorbed vinyl system. As such, it is unclear to what extent a single potential would be crossed at which this state would suddenly become occupied, and more likely that pushing the Cu electrode negatively progressively increases the occupation of these states. Without DFT consideration of what is speculated as the distinguishing mechanistic feature of the work, the DFT work in this manuscript is collectively adding more confusion than useful addition to the study.

We thank the reviewer for their comment. First of all, although both Raman and IR spectra show possible adsorption and accumulation of vinyl alcohol on the Cu surface, it is hard to measure the adsorption strength of vinyl alcohol in these measurements. For example, perchlorate is usually considered to be a non-specifically adsorbed anion (<https://doi.org/10.1038/s41929-022-00810-6>) on clean metal electrode surface which means the interaction between perchlorate and metal surface is extremely weak (<https://doi.org/10.1038/s42004-022-00635-1>). However, the stretching signal of uncoordinated perchlorate located at 931 cm^{-1} is commonly detected from the Raman spectra near the electrode surface region. Thus, the strength of the interaction of the vinyl alcohol to the electrode surface is uncertain.

According to our calculated potential dependent vinyl alcohol adsorption free energy, shown in Fig. R4 ©, this adsorption is endergonic (ca. 0.1 eV) on Cu(100) terrace site and slightly exergonic (ca. -0.2 eV) on Cu(211) step site within the potential window of $-0.7 V_{\text{RHE}}$ to $+0.7 V_{\text{RHE}}$. So, it is possible that adsorption of vinyl alcohol is rather weak and is only favorable on step sites on Cu. Under this condition, it would make sense that a specific potential would be required to transfer one electron to the unoccupied state of the vinyl alcohol to form a charged intermediate. As shown in Fig. R13, this assumption is possibly true because we observed a similar onset potential of $-0.7 V_{\text{RHE}}$ for both Cu disk and Cu powder electrode. This suggests that the onset potential is not influenced by surface morphology. Moreover, the magnitude of ethylene/ethane oxidation current is much higher on Cu powder compared to those formed on a Cu disk. This may be due to the higher surface area on Cu powder providing more active sites. However, this surface morphology difference does not influence the onset potential for reducing adsorbed vinyl alcohol to ethylene and ethane on polycrystalline Cu.

Fig. R13 | Control experiment by using Cu powder as working electrode. Right: Cyclic voltammograms (CVs) of products detected by the Pt ring electrode during Cu powder catalyzed acetaldehyde (100 mM) reduction at pH = 7 (0.1 M NaClO₄). Only the positive-going scans were shown for clarity. Cu powder potential was held from -0.5 V to -1.0 V. CV of the acetaldehyde blank was collected on the Pt ring when Cu powder was at open circuit potential (OCP). Left: Total Cu powder current during acetaldehyde reduction. CV conditions: 0 to 1.6 V and then back to 0 V (vs. RHE), 100 mV/s, 1600 RPM, 298 K.

Second, CO dimerization was also believed to proceed via a decoupled electron transfer to the adsorbed neighboring CO on Cu. This involves a C-C coupling step mediated by an electron transfer to the adsorbed CO to form C₂O₂⁻ intermediate. Here, the proton transfer happens only after the formation of a negatively charged C₂O₂⁻ intermediate on the surface ([10.1002/anie.201301470](https://doi.org/10.1002/anie.201301470); [10.1021/acs.jpcllett.5b00722](https://doi.org/10.1021/acs.jpcllett.5b00722); <https://doi.org/10.1021/acs.jpcllett.5b01559>; <https://doi.org/10.1038/s41467-021-23582-2>). More importantly, previous electrochemical reduction of CO show that C₂O₂²⁻ can be produced in non-aqueous media on Ni, Pt and Hg electrodes which suggests that C-C coupling indeed involves electron transfer to the adsorbed CO without any proton transfer

([10.1016/S0022-0728\(83\)80042-4](https://doi.org/10.1016/S0022-0728(83)80042-4)). Therefore, we think our proposed decoupled electron transfer to the adsorbed vinyl alcohol mechanism is plausible. However, advanced computational methodology is needed to gain more insight into this key step in the future.

4) The discussion of the role of mass transfer and solution phase tautomerization to the enol in dictating the reduction paths is rather nebulous, and hard to allocate then to the mechanism proposed. It appears the authors are assuming all acetaldehyde/vinyl alcohol conversion would occur far from the surface, rather than the adsorbed vinyl alcohol species formation being part of a surface catalyzed reaction path. The basis for this assumption should be stated, as it is unclear why the opposite (ie, Cu promoting vinyl alcohol formation through surface reaction processes due to its favorable formation) could not be assumed. Further, the discussion of mass transfer effects appears to assume the tautomerization in solution is much slower than the transport of acetaldehyde or the enol to the surface, without providing explanation for this assumption.

We thank the reviewer for their comment. The reviewer is correct that we think that the acetaldehyde/vinyl alcohol conversion occurs far from the surface. In other word, acetaldehyde and its enol tautomer is at equilibrium in the bulk aqueous solution. We do not think that vinyl alcohol formation from acetaldehyde can be catalyzed by Cu for three reasons: First, acetaldehyde was reported to be a non-adsorbed species on Cu surface during typical electroreduction condition on Cu (<https://doi.org/10.1002/anie.201508851>; <https://doi.org/10.1016/j.cattod.2015.09.029>). This is consistent with our in-situ spectroscopic results where no surface adsorbed acetaldehyde signal can be detected. Second, if acetaldehyde is catalyzed on the surface to form vinyl alcohol, we should also see ethylene and ethane formation under static electrochemical conditions. This is not consistent with our observation where the formation of ethylene and ethane is very sensitive to mass transport condition. Third, our RRDE and RDE-GC-MS experiment results for 0.1 M acetone and 0.1 M propionaldehyde show that the hydrocarbons resulted from the reduction of corresponding enols highly depend on the pK_e values of the ketone-enol tautomerization reaction. Briefly, hydrocarbons can be clearly detected from acetaldehyde and propionaldehyde reduction but not for acetone under high reducing electrode potential. Remember that $pK_e(\text{acetone}) \ll pK_e(\text{acetaldehyde}) < pK_e(\text{propionaldehyde})$. These pK_e values resembles the stability of enols of the corresponding ketones and aldehydes in bulk solution. Thus, we believe that all the acetaldehyde/vinyl alcohol conversion happens in the bulk solution, not catalyzed on the Cu surface.

We believe that the mass transfer effect is of great importance when considering extremely low concentration reactive species such as vinyl alcohol in this study. However, we make no assumption of the rate of tautomerization. Meaning, we do not assume that the tautomerism is occurring at a much slower rate than the transport of acetaldehyde or the enol to the surface. Moreover, we ascribe the enhanced ethylene and ethane production at high electrode potential to a faster mass transport effect by increasing the rotation rate of the electrode while assuming mass transport effect has negligible influence over keto-enol equilibrium, that is, equilibrium constant does not change as we change the rotation rate of the electrode. For an ideal mass transport limited condition in RDE setup, the diffusion layer thickness at steady state can be estimated according to the Levich equation (Bard, A. J., &

Faulkner, L. R. (2000). *Electrochemical methods and applications*. New York: Wiley-Interscience):

$$\delta_F = 1.61 D_F^{\frac{1}{3}} \nu^{\frac{1}{6}} \omega^{-\frac{1}{2}}$$

where D_F is the diffusion coefficient of the reactant, ν is the kinematic viscosity of the solution and ω is the rotation rate. At a high rotation rate (≥ 1600 RPM), the diffusion layer thickness is around $1\mu\text{m}$ which is a few orders of magnitude smaller than those experience in static condition. (<https://doi.org/10.1021/acscatal.1c00272>; <https://doi.org/10.1021/jacs.9b10061>; <https://doi.org/10.1021/jacs.8b04058>)

One would expect that an extreme dilute species such as vinyl alcohol will be very sensitive to mass transport. This is indeed correlated to ethylene and ethane formation observed in this work where increasing the rotation rate is directly proportional to the products observed. We also believe that ethylene and ethane were not observed and reported in previous studies on the electroreduction of acetaldehyde due to the slow mass transport typical in the most common electrochemical cell used in these studies.

The following statement is added to the main text (line 415 -418, page 23) to improve clarity.

..as we assume that tautomerization is not catalyzed by the Cu surface due to acetaldehyde doesn't specifically adsorb on the Cu surface under typical electroreduction condition(7, 8) and hydrocarbons can't be detected under static condition in our experiment..

REVIEWER COMMENTS

Reviewer #1 (Remarks to the Author):

The authors have carried out additional experiments, added relevant citations, and addressed the issues raised by the reviewers. The questions raised by the reviewers are common problems in recent CO₂ electrolysis research. 1) Gaps often exist between performance testing and spectroscopy studies. The device design and test conditions for spectroscopy are often different from those used for performance testing. In particular, the structure of the RRDE makes it difficult to achieve SEIRAS measurements under working conditions. While Raman analysis may be possible using RRDE, the electrolyzer needs to be specifically designed to avoid electrolyte leakage and bubble adhesion on the electrode surface. 2) DFT calculations for electrochemistry require a more careful simulation of the electrochemical interface and a deeper understanding of the structure of the electrochemical double layer. DFT calculations incorporating solvation effects (water, cations, and anions) are also an important and frontier research area. In summary, these problems exist not only for this work but also for many other published works. Although it is a pity that the authors were not more successful in establishing a closer correspondence among the RRDE testing, spectral characterizations, and DFT calculations, the present results can still shed light on a new reaction mechanism from a qualitative point of view (ethylene and ethane can be obtained from adsorbed vinyl alcohols). I appreciate the efforts that authors have made. Thus, I recommend this work for publication with minor revisions, which is expected to inspire new discussions among researchers on the C₂⁺ formation mechanisms of Cu-based catalysts.

Major comments:

1. The authors performed Raman tests using D₂O as solvent. What are the variations of the peaks except for the peaks related -CH₃ (-CD₃) and -CH₂ (-CD₂)? Can the authors show the full-range Raman spectra and analyze them?
2. The authors give Raman spectra from -0.3 to -0.9 V vs. RHE. The results are a little strange in that the peak intensity hardly changes with the potential. If vinyl alcohol is the intermediate that produces ethylene and ethane, why does the corresponding peak area (intensity) of vinyl alcohol not decrease as the potential becomes negative?
3. The authors give the reason for not observing a significant Stark tuning effect. According to the literature (DOI: 10.1021/jacs.9b11817), the shifts of the peak corresponding to C-O vibration are observable, thus why is it not observed?
4. Can the authors give information on the XRD pattern of Cu powder and polycrystalline Cu? Even for commercial Cu, samples from different manufacturers often present different structures (10.1002/cctc.202200540). Authors should try to ensure that the Cu used for RRDE testing and spectroscopy measurements shares a similar structure.
5. According to the schematic diagram (ATR-SEIRAS) of the device provided by the authors, I do not observe the two chambers of the reactor. Therefore, the authors need to be extra careful about the interference of the Pt counter electrode on the spectra (DOI: 10.1021/acs.jpcc.8b05634). In addition,

according to the reactor structure provided by the authors, I cannot find any obvious inlet and outlet liquid pipelines (DOI: 10.1021/acscatal.0c03553), how did the authors achieve the electrolyte flow (15 mL/min)?

6. As described by the authors, the RRDE-GC-MS suffers from gas leakage. Could it be that the H₂ was not detected because of gas leakage from the device?

Reviewer #2 (Remarks to the Author):

This reviewer is pleased to see that the manuscript is improved as compared to the last version. However, the revised manuscript still lacks some decisive evidences to verify the reaction mechanism of acetaldehyde reduction on Cu. The following lists some suggestions for the authors to further improve this manuscript to meet the high standard of publication in Nature Communications:

1. As for the concern on the local pH, the complementary data in the latest version of the manuscript have convinced me that the deviation of local pH will not lead to drastic change of reaction mechanism of acetaldehyde reduction on Cu electrode. Although the local pH may change the competition between HER and acetaldehyde reduction kinetics, the choice of perchlorate as electrolyte is mostly acceptable in this study. However, one of the main conclusions is that “the RDS step is more likely to be an electron transfer step rather than commonly assumed thermochemical step or CPET step”. And this statement is inferred from the experimental observation that “the onset potentials for the formation of ethane and ethylene were found to be pH independent on the SHE scale”. To be prudent, the authors are advised to measure the onset potentials for the formation of ethane and ethylene at pH 4, 7, and 10 in phosphate buffer solution to double-check this core experimental evidence.

2. As for the Raman spectroscopy, the reviewer is glad to see the neat and sharp C-D stretching band in the H/D isotopic labelling experiment as shown in Fig. R 16, raising the possibility to obtain more concrete evidences for reaction mechanism analysis. Note that acetaldehyde coexists in equilibrium with its isomer enol. And the H/D exchange of protonic H/D is kinetically fast. Therefore, in D₂O, it is expected that H/D exchange between OH group of enol with heavy water will lead to a nearly complete deuteration of OH group of enol in the heavy water solution. Eventually, limited by the slow rate of isomerization of acetaldehyde to enol, a deuteration of methyl group (only CH₃) of acetaldehyde will be accomplished after a long period of time. On the other hand, as shown by the authors, the formation of adsorbed H/D on Cu may cause surface H/D exchange between adsorbed organic species and adsorbed H/D, leading to deuteration of both CH₂ and CH₃ of the intermediate CH₂CH₃O*. Therefore, the authors are suggested to either provide concrete evidences or design control experiments for precise assignment of the C-D band between 2100-2200 cm⁻¹ as shown in Fig. R 16. If this band is exclusively assigned to CD₃, it can be inferred that Cu is capable of catalyzing the isomerization between acetaldehyde and enol at -0.3 V; If this band is assigned to CD₂ and CD₃, it can be inferred that Cu can catalyze the reduction of enol to ethoxy intermediate at -0.3 V.

3. This reviewer also notices that the authors have not analyzed the in-situ Raman spectra in phosphate buffer solution. The bands between 400 and 1200 cm^{-1} were indiscriminately assigned to adsorbed phosphate species by the authors. As I can see, these bands get stronger against the negatively shifted electrode potential. However, the adsorption of phosphate species is unfavorable with negatively going potentials, therefore the bands for phosphate species should have turned weaker accordingly, as previously reported in <https://doi.org/10.1021/jp970097k>. So, these strengthened Raman bands should not be assigned to adsorbed phosphate species. The authors are suggested to carefully analyze these valuable spectral data.

4. Supplementary Figure 20 (in-situ Raman spectra between -0.4 to -0.9 V) should be merged with Figure 3.a (in-situ Raman spectra between 0.3 to -0.3 V) and presented in the main text. The information in Supplementary Figure 20 is indispensable to conclude that “vinyl alcohol and ethoxy intermediate present on Cu surface during the reduction of acetaldehyde”. In addition, the authors are suggested to acquire the in-situ Raman spectra beyond -1.2 V where the formation of ethane and ethylene starts. Due to the restricted mass transport of a reactant from solution, the surface vinyl alcohol signal is expected to vanish in the Raman spectra against negatively shifted potential. This expected observation will be decisive to conclude that the precursor of ethane and ethylene is the surface vinyl alcohol, and the surface vinyl alcohol is limited by the mass transport from the bulk solution. If the Raman band of surface vinyl alcohol still intensifies beyond -1.2 V, a specific explanation of the potential dependent band intensity should be given, otherwise, the kinetic model given by the reaction mechanism may be questionable.

5. As for the Question 4 of Reviewer 3, the authors should note that there are four possible kinetic models that should be separately discussed. (1) For model 1, tautomerization in solution is much slower than the transport of acetaldehyde or enol to the surface, and acetaldehyde is the direct reaction precursor, it undergoes a catalytic step to be transformed into adsorbed enol on Cu. In this case, there will be a concentration gradient of acetaldehyde in the diffusion layer, whereas the enol concentration is constant in the diffusion layer. (2) For model 2, tautomerization in solution is even faster than the transport of acetaldehyde or the enol to the surface, and acetaldehyde is the reaction precursor, it undergoes a catalytic step to be transformed into adsorbed enol on Cu. In this case, there will be a concentration gradient of both enol and acetaldehyde in the diffusion layer. (3) For model 3, tautomerization in solution is much slower than the transport of acetaldehyde or the enol to the surface, and enol is the reaction precursor. In this case, there will be a concentration gradient of enol in the diffusion layer, whereas acetaldehyde concentration is constant in the diffusion layer. (4) For model 4, tautomerization in solution is even faster than the transport of acetaldehyde or the enol to the surface, and enol is the reaction precursor. In this case, there will be concentration gradients of both enol and acetaldehyde in the diffusion layer.

In model 1 or 3, only the direct precursor (either acetaldehyde or enol) contributes to the reaction current, whereas in model 2 or 4, due to the fast tautomerization in solution, both enol and acetaldehyde contribute to the reaction current. Bear these in mind, we can infer that for each model, there will be a mass-transport limiting current when the surface concentration of the reaction precursor drop to 0 at RDE mode. The current value is expected to be proportional to the gradient of the corresponding reactant concentration (acetaldehyde concentration for model 1, enol concentration for

model 3, and sum concentration for models 2 and 4), the authors are suggested to evaluate the mass-transport limiting current values for different models and compare them with the measured values to determine the true kinetic model.

Reviewer #3 (Remarks to the Author):

The authors have well addressed all points I brought up in review, as well as those of the other reviewers. Revision has either clarified concerns (especially with respect to assignments of oxidation features on Pt to ethylene and ethane) or made clear limitations of the work so they are apparent to all readers. As such, I recommend publication of the revised work. I still believe this is a stronger contribution if the DFT portions of the study were removed, as the limitations of the approaches used lead to mechanistic conclusions being better supported by experimental data, with the DFT portions added more confusion due to these limitations than usefulness in supporting conclusions. However, these limitations are apparent to a knowledgeable reader, and the DFT work is not misrepresented with limitations discussed, if the authors still want to include this portion of their collaborative work.

Again, we thank the reviewers for their constructive comments. These comments have helped us improve the quality of our manuscript significantly. In this revision, we have included additional experiments and explanation to address the reviewers' concerns and suggestions.

Reviewer #1 (Remarks to the Author):

The authors have carried out additional experiments, added relevant citations, and addressed the issues raised by the reviewers. The questions raised by the reviewers are common problems in recent CO₂ electrolysis research. 1) Gaps often exist between performance testing and spectroscopy studies. The device design and test conditions for spectroscopy are often different from those used for performance testing. In particular, the structure of the RRDE makes it difficult to achieve SEIRAS measurements under working conditions. While Raman analysis may be possible using RRDE, the electrolyzer needs to be specifically designed to avoid electrolyte leakage and bubble adhesion on the electrode surface. 2) DFT calculations for electrochemistry require a more careful simulation of the electrochemical interface and a deeper understanding of the structure of the electrochemical double layer. DFT calculations incorporating solvation effects (water, cations, and anions) are also an important and frontier research area. In summary, these problems exist not only for this work but also for many other published works. Although it is a pity that the authors were not more successful in establishing a closer correspondence among the RRDE testing, spectral characterizations, and DFT calculations, the present results can still shed light on a new reaction mechanism from a qualitative point of view (ethylene and ethane can be obtained from adsorbed vinyl alcohols). I appreciate the efforts that authors have made. Thus, I recommend this work for publication with minor revisions, which is expected to inspire new discussions among researchers on the C₂⁺ formation mechanisms of Cu-based catalysts.

Major comments:

1. The authors performed Raman tests using D₂O as solvent. What are the variations of the peaks except for the peaks related -CH₃ (-CD₃) and -CH₂ (-CD₂)? Can the authors show the full-range Raman spectra and analyze them?

We thank the reviewer for this helpful comment, we have extended the x-axis of our original Raman to present the spectra with a broader range (600 to 3200 cm⁻¹) as shown in **Fig.R1**. In D₂O based electrolyte, we only observe band variations related to -CH₃(CD₃) and CH₂(CD₂). In the previous reply, we assigned two bands at 2183 cm⁻¹ and 2111 cm⁻¹ at -0.3 V to symmetrical stretching of -CD₃ and -CD₂ groups (<http://dx.doi.org/10.1139/v56-178>; <https://doi.org/10.1063/1.456273>) from adsorbed ethoxy intermediates, respectively. Moreover, these assignments can be further supported by Raman bands located at lower frequency region. As highlighted in the yellow blocked area where two bands appeared at 885 cm⁻¹ and 847 cm⁻¹ at -0.3 V in D₂O can be assigned to the rocking vibration of CD₃ group

(<https://doi.org/10.1515/zpch-2018-1368>). The band at 636 cm^{-1} at -0.3 V in D_2O is assigned to the rocking vibration of CD_2 group (<https://doi.org/10.1021/bi00099a010>; [https://doi.org/10.1016/0584-8539\(67\)80007-2](https://doi.org/10.1016/0584-8539(67)80007-2); [https://doi.org/10.1016/S0584-8539\(87\)80057-0](https://doi.org/10.1016/S0584-8539(87)80057-0)).

Fig.R1 | Additional in situ Raman spectra obtained on polycrystalline copper powder electrode as a function of potential in Ar saturated 0.1 M NaClO_4 ($\text{pH}=7$) with 100 mM acetaldehyde. The electrolyte is prepared in H_2O (left) and D_2O (right).

We added the following discussion to the Supplementary Information as Supplementary Note 1:

To make our band assignments more reliable, in situ Raman were conducted in D_2O based electrolyte solution as shown in Supplementary Fig.22. Two bands appeared at 2183 cm^{-1} and 2111 cm^{-1} at -0.3 V in D_2O which are assigned to the symmetrical C-D stretching of CD_3 and CD_2 groups of the adsorbed ethoxy intermediate^{1,2}. These assignments can be further supported by Raman bands located at lower frequency region. As highlighted in the yellow region, two bands appeared at 885 cm^{-1} and 847 cm^{-1} at -0.3 V in D_2O can be assigned to the rocking vibration of CD_3 group⁴. The band appeared at 636 cm^{-1} at -0.3 V in D_2O is assigned to the rocking vibration of CD_2 group⁵⁻⁷. These groups may have been formed via a surface H/D exchange process as reported in a previous study⁸. Collectively, the shifts in the Raman bands supports the assignment of $2800\text{-}3000\text{ cm}^{-1}$ bands to adsorbed ethoxy, as opposed to the presence of an exogenous species.

More importantly, OH group of vinyl alcohol will be completely deuterated if we assume a fast H/D exchange between the OH group of vinyl alcohol and D₂O, considering that tautomerization of acetaldehyde to vinyl alcohol is slow then it's possible that even adsorbed vinyl alcohol can be reduced to ethoxy on Cu surface at -0.3 V due to the bands between 2100-2200 cm⁻¹ are assigned to CD₂ and CD₃ groups at -0.3 V in D₂O. However, more well-designed experiments should be conducted to further support this argument and our future work will focus on this.

2. The authors give Raman spectra from -0.3 to -0.9 V vs. RHE. The results are a little strange in that the peak intensity hardly changes with the potential. If vinyl alcohol is the intermediate that produces ethylene and ethane, why does the corresponding peak area (intensity) of vinyl alcohol not decrease as the potential becomes negative?

We appreciate this comment from the reviewer. To observe the peak intensity, change as a function of applied potential, we expanded the high frequency region (2700 cm⁻¹ to 3100 cm⁻¹) of our original Raman spectra as shown in **Fig.R2**. We observe that the band around 3045 cm⁻¹ decreases from -0.6 V_{RHE}, which was previously assigned to =C-H stretching of adsorbed vinyl alcohol, this result is in reasonable agreement with our RRDE measured onset potential (ca. -0.7 V_{RHE}) for reducing vinyl alcohol to ethylene and ethane on Cu powder electrode. Moreover, we observed an opposite trend for -CH₃ and -CH₂ between 2800 cm⁻¹ and 3000 cm⁻¹ as these bands tend to increase as the potential is stepped to values negative of -0.6 V_{RHE}, possibly due to the accumulation of reduction products near the electrode surface at high reducing potentials. We have now included a discussion of this observation into the main manuscript, which is also copied below:

... Furthermore, it is observed that this band tends to decrease as potential becomes more negative than -0.6 V, which is in reasonable agreement with our RRDE measured onset potential (ca. -0.7 V) for reducing vinyl alcohol to ethylene and ethane on Cu powder electrode as shown in Supplementary Fig.5....

Fig.R2 | In situ Raman spectra obtained on polycrystalline copper as a function of potentials in Ar saturated 0.1 M NaClO₄ (pH=7) with 100 mM acetaldehyde. All the potentials are reference to the RHE scale.

3. The authors give the reason for not observing a significant Stark tuning effect. According to the literature (DOI: 10.1021/jacs.9b11817), the shifts of the peak corresponding to C-O vibration are observable, thus why is it not observed?

We thank the reviewer for this comment. We want to clarify that while Stark tuning effect is not evident in the full spectra range (600-3100 cm⁻¹), it is observed for C-O. We assigned the band around 1005 cm⁻¹ to C-O stretching vibration of adsorbed ethoxy intermediate. Fig. R3 is an expanded version of our original spectra where we focus on the region where C-O stretching is observed. As suggested in the literature (DOI: 10.1021/jacs.9b11817), Chang et al. observed a red-shift of C-O vibration peak as OD-Cu electrode potential becomes more negative by using in-situ SEIRAS. As shown in **Fig.R3**, we also observe a similar C-O vibration shift trend from 1010 cm⁻¹ to 1002 cm⁻¹ as our electrode potential changes from 0.04 V to -0.3 V. We included a discussion of this in the main manuscript, which is also copied below:

...More importantly, we observe that C-O vibration band shifts to a lower frequency as potential becomes more negative due to the Stark tuning effect as shown in Supplementary Fig.22, which is consistent with a previous study reported by Chang et al. that a red-shift of C-O vibration was observed on OD-Cu electrode as potential becomes more negative (11)...

Fig.R3 | In situ Raman spectra around C-O stretching vibration region obtained on polycrystalline copper as a function of potential in Ar saturated 0.1 M NaClO₄ (pH=7) with 100 mM acetaldehyde. All the potentials are reference to the RHE scale.

4. Can the authors give information on the XRD pattern of Cu powder and polycrystalline Cu? Even for commercial Cu, samples from different manufacturers often present different structures (10.1002/cctc.202200540). Authors should try to ensure that the Cu used for RRDE testing and spectroscopy measurements shares a similar structure.

We thank the reviewer for this helpful comment, the XRD patterns of Cu powder and polycrystalline Cu disks are shown in **Fig.R4**. All major Cu diffraction peaks are present with no additional or peaks resulting from impurities were observed. The highest peak intensity was observed for Cu(111) around 43.5°, Cu(100) corresponds to a medium peak intensity at 50.5° and lowest peak intensity was observed for Cu(110) around 74.2°. And we found that Cu(110) signal was almost negligible in RDE-Cu and RRDE-Cu, which was also observed for some commercial polycrystalline copper foils provided by several suppliers in the literature as reviewer mentioned (10.1002/cctc.202200540). However, this difference in the ratio of the crystal orientation was not observed to influence the onset potential for the formation of ethylene and ethane according to our RRDE experiments, the onset potentials were determined to be $-0.75 V_{RHE}$ and while we observe $-0.7 V_{RHE}$ on RRDE-Cu disk electrode and Cu powder electrode, respectively. We think that single crystal study will provide more insight into understanding the relation between reaction activity and surface orientation of Cu, which can be the focus of our future work. This result has been added to supplementary Figure 30.

Fig.R4 | X-ray diffraction (XRD) patterns of polycrystalline Cu powder (yellow), RDE-Cu disk (green) and RRDE-Cu disk (red). XRD data was collected on a Bruker D8 Advance A25 diffractometer using a Cu K_{α} X-ray tube.

5. According to the schematic diagram (ATR-SEIRAS) of the device provided by the authors, I do not observe the two chambers of the reactor. Therefore, the authors need to be extra careful about the interference of the Pt counter electrode on the spectra (DOI: 10.1021/acs.jpcc.8b05634). In addition, according to the reactor structure provided by the authors, I cannot find any obvious inlet and outlet liquid pipelines (DOI: 10.1021/acscatal.0c03553), how did the authors achieve the electrolyte flow (15 mL/min)?

We thank the reviewer for this comment. Our anode chamber and cathode chamber were separated by a Nafion membrane as shown in Fig. R5. We agree with reviewer that rigorous spectro-electrochemical studies should use high purity electrolyte after pre-electrolysis, membrane separator and high purity graphite counter electrode or a CE that will not alter the observations on the WE. Here, our CE was in a separate chamber with a proton exchange membrane isolating the catholyte from the anolyte. We have added this detail and this reference ([10.1021/acs.jpcc.8b05634](https://doi.org/10.1021/acs.jpcc.8b05634)) to the experimental section in the main text, which is also copied below:

...Pt foil was used as the counter electrode and reference electrode was the same as that used in Raman tests. Anode chamber and cathode chamber were separated by Nafion membrane to prevent the interference from Pt CE reported by Dunwell et al. (55)...

Fig.R5 | Schematic of anode chamber used in ATR-SEIRAS tests.

Our apologies for the confusion, we want to clarify that our electrolyte flow rate of (15 mL/min) was used in the Raman experiments, but the electrolyte for the ATR-SEIRAS was not flowed. We have added additional descriptions to the experimental section of ATR-SEIRAS to avoid bringing any confusion to the readers.

6. As described by the authors, the RRDE-GC-MS suffers from gas leakage. Could it be that the H₂ was not detected because of gas leakage from the device?

We thank the reviewer for this comment. We cannot exclude the possibility that gas leakage is the reason for not observing H₂ signal in the GCMS at low HER rates. Since the amount of H₂ generated could possibly be under the detection limit of our GCMS. As a result, we think both HER activity and gas leakage should be considered in explaining the absence of H₂ signal in this work.

Reviewer #2 (Remarks to the Author):

This reviewer is pleased to see that the manuscript is improved as compared to the last version. However, the revised manuscript still lacks some decisive evidences to verify the reaction mechanism of acetaldehyde reduction on Cu. The following lists some suggestions for the authors to further improve this manuscript to meet the high standard of publication in Nature Communications:

1. As for the concern on the local pH, the complementary data in the latest version of the manuscript have convinced me that the deviation of local pH will not lead to drastic change of reaction mechanism of acetaldehyde reduction on Cu electrode. Although the local pH may change the competition between HER and acetaldehyde

reduction kinetics, the choice of perchlorate as electrolyte is mostly acceptable in this study. However, one of the main conclusions is that “the RDS step is more likely to be an electron transfer step rather than commonly assumed thermochemical step or CPET step”. And this statement is inferred from the experimental observation that “the onset potentials for the formation of ethane and ethylene were found to be pH independent on the SHE scale”. To be prudent, the authors are advised to measure the onset potentials for the formation of ethane and ethylene at pH 4, 7, and 10 in phosphate buffer solution to double-check this core experimental evidence.

We thank the reviewer for this helpful suggestion, in this revision, we have included RRDE experiments pH 4, 7 and 10 in 0.1 M phosphate solution as shown in **Fig.R6**. The onset potentials for the formation of ethylene and ethane are $-0.9 V_{RHE}$, $-0.8 V_{RHE}$ and $-0.6 V_{RHE}$ at pH = 4, 7 and 10, respectively. These results are in good agreement with the experimental results acquired from perchlorate solution, which are $-0.9 V_{RHE}$, $-0.75 V_{RHE}$ and $-0.5 V_{RHE}$ at pH = 4, 7 and 10, respectively. The onset potential observed in the phosphate solution indicates that the reaction is not dependent on the solution pH. Thus, we tend to believe that the RDS step for the formation of ethylene and ethane is possible an electron transfer step. This set of observations is now included in supplementary figure 4.

Fig.R6| Control experiment in 0.1 M phosphate buffer solution. a,b,c Cyclic voltammograms (CVs) of products detected by the Pt ring electrode during Cu catalyzed acetaldehyde reduction at pH = 4, 7, 10, respectively. Only the positive-going scans were shown for clarity. CV of the acetaldehyde blank was collected on the Pt ring when Cu disk was at open circuit potential (OCP). **d,e,f** Total Cu disk current during acetaldehyde reduction. CV conditions: 0 to 1.6 V and then back to 0 V (vs. RHE), 100 mV/s, 1600 RPM, 298 K.

2. As for the Raman spectroscopy, the reviewer is glad to see the neat and sharp C-D stretching band in the H/D isotopic labelling experiment as shown in Fig. R 16, raising the possibility to obtain more concrete evidences for reaction mechanism analysis. Note that acetaldehyde coexists in equilibrium with its isomer enol. And the H/D exchange of protonic H/D is kinetically fast. Therefore, in D₂O, it is expected that H/D exchange between OH group of enol with heavy water will lead to a nearly

complete deuteration of OH group of enol in the heavy water solution. Eventually, limited by the slow rate of isomerization of acetaldehyde to enol, a deuteration of methyl group (only CH₃) of acetaldehyde will be accomplished after a long period of time. On the other hand, as shown by the authors, the formation of adsorbed H/D on Cu may cause surface H/D exchange between adsorbed organic species and adsorbed H/D, leading to deuteration of both CH₂ and CH₃ of the intermediate CH₂CH₃O*.

Therefore, the authors are suggested to either provide concrete evidences or design control experiments for precise assignment of the C-D band between 2100-2200 cm⁻¹ as shown in Fig. R 16. If this band is exclusively assigned to CD₃, it can be inferred that Cu is capable of catalyzing the isomerization between acetaldehyde and enol at -0.3 V; If this band is assigned to CD₂ and CD₃, it can be inferred that Cu can catalyze the reduction of enol to ethoxy intermediate at -0.3 V.

We thank the reviewer for this comment, as shown in **Fig. R1**, extended the x-axis of our original Raman spectra to show a broader range to make our band assignments to -CD₂ and -CD₃ more reliable. In the previous reply, we assigned two bands at 2183 cm⁻¹ and 2111 cm⁻¹ at -0.3 V to symmetrical stretching of -CD₃ and -CD₂ groups (<http://dx.doi.org/10.1139/v56-178>; <https://doi.org/10.1063/1.456273>) from adsorbed ethoxy intermediates, respectively. These assignments can be further supported by Raman bands located at lower frequency region. As highlighted in yellow block region, two bands appeared at 885 cm⁻¹ and 847 cm⁻¹ at -0.3 V in D₂O can be assigned to the rocking vibration of CD₃ group (<https://doi.org/10.1515/zpch-2018-1368>). The band at 636 cm⁻¹ at -0.3 V in D₂O is assigned to the rocking vibration of CD₂ group ([https://doi.org/10.1016/0584-8539\(67\)80007-2](https://doi.org/10.1016/0584-8539(67)80007-2); <https://doi.org/10.1021/bi00099a010>; [https://doi.org/10.1016/S0584-8539\(87\)80057-0](https://doi.org/10.1016/S0584-8539(87)80057-0)). According to these assignments, we tend to believe that the bands between 2100 to 2200 cm⁻¹ should be assigned to -CD₂ and -CD₃ of adsorbed ethoxy. Thus, as reviewer suggested, it is possible that Cu surface can catalyze the reduction of vinyl alcohol to ethoxy intermediate at -0.3 V_{RHE} on Cu, and ethoxy can be further reduced to ethanol. We believe this result is important and unexpected so we decide to add corresponding discussion to Supplementary Note 1, which is also copied below:

To make our band assignments more reliable, in situ Raman tests were conducted in D₂O based electrolyte solution as shown in Supplementary Fig.22. Two bands appeared at 2183 cm⁻¹ and 2111 cm⁻¹ at -0.3 V in D₂O which are assigned to the symmetrical C-D stretching of CD₃ and CD₂ groups of the adsorbed ethoxy intermediate ^{1,2}. These assignments can be further supported by Raman bands located at lower frequency region. As highlighted in yellow color region, two bands appeared at 885 cm⁻¹ and 847 cm⁻¹ at -0.3 V in D₂O can be assigned to the rocking vibration of CD₃ group ⁴. The band appeared at 636 cm⁻¹ at -0.3 V in D₂O is assigned to the rocking vibration of CD₂ group ⁵⁻⁷. These groups may have been formed via a surface H/D exchange process as reported in a previous study ⁸. Collectively, the shifts in the Raman bands supports the assignment of 2800-3000 cm⁻¹ bands to

adsorbed ethoxy, as opposed to the presence of an exogenous species.

More importantly, OH group of vinyl alcohol will be completely deuterated if we assume a fast H/D exchange between the OH group of vinyl alcohol and D₂O, considering that tautomerization of acetaldehyde to vinyl alcohol is slow then it's possible that even adsorbed vinyl alcohol can be reduced to ethoxy on Cu surface at -0.3 V due to the bands between 2100-2200 cm⁻¹ are assigned to CD₂ and CD₃ groups at -0.3 V in D₂O. However, more well-designed experiments should be conducted to further support this argument and our future work will focus on this.

Fig.R1 | Additional in situ Raman spectra obtained on polycrystalline copper powder electrode as a function of potentials in Argon saturated 0.1 M NaClO₄ (pH=7) with 100 mM acetaldehyde. The electrolyte is based on H₂O(left) and D₂O(right).

3. This reviewer also notices that the authors have not analyzed the in-situ Raman spectra in phosphate buffer solution. The bands between 400 and 1200 cm⁻¹ were indiscriminately assigned to adsorbed phosphate species by the authors. As I can see, these bands get stronger against the negatively shifted electrode potential. However, the adsorption of phosphate species is unfavorable with negatively going potentials, therefore the bands for phosphate species should have turned weaker accordingly, as previously reported in <https://doi.org/10.1021/jp970097k>. So, these strengthened Raman bands should not be assigned to adsorbed phosphate species. The authors are suggested to carefully analyze these valuable spectral data.

We thank the reviewer for this comment. As shown in **Fig.R7**, we have now labelled each band between 500 and 1200 cm⁻¹ to exclude the interference of the bands

resulting from adsorbed phosphates. According to the previous literature, the reviewer mentioned (<https://doi.org/10.1021/jp970097k>) and other relevant published work (<https://doi.org/10.1021/jp035594h>; <https://doi.org/10.1021/acs.jpcclett.6b01342>), the small band around 635 cm^{-1} can be assigned to asymmetrical stretching vibration from PO_4^{3-} . Another small band at 717 cm^{-1} is assigned to stretching vibration from H_2PO_4^- species. It is important to note that primary phosphate adsorption band is located at 936 cm^{-1} , which can be assigned to P-O^* (O^* denotes adsorbed oxygen) stretching vibration from adsorbed PO_4^{3-} , HPO_4^{2-} or H_2PO_4^- . It is possible that P-O stretching vibration band resulted from H_2PO_4^- overlaps with C-O stretching band of ethoxy at 1018 cm^{-1} , so we assign the band at 1018 cm^{-1} to both ethoxy and phosphate.

Fig.R7| Control experiment for in-situ Raman experiment in 0.1 M phosphate buffer solution (pH = 7) at indicated conditions.

In this revision, we updated this figure and included additional discussion to the supplementary note 2, which is also copied below:

According to a previous surface enhanced Raman spectroscopy (SERS) study on phosphate anions adsorption on Cu electrode⁹, each band between 500 and 1200 cm^{-1} was labeled to differentiate the bands resulted from adsorbed phosphates and ethoxy. The band around 635 cm^{-1} can be assigned to asymmetrical stretching vibration from PO_4^{3-} . The small band at 717 cm^{-1} is assigned to stretching vibration from H_2PO_4^- species. It's important to note that primary phosphate adsorption band is located at 936 cm^{-1} , which can be assigned to P-O^ (O^* denotes adsorbed oxygen) stretching vibration from adsorbed PO_4^{3-} , HPO_4^{2-} or H_2PO_4^- . It's possible that P-O stretching vibration band resulted from H_2PO_4^- overlaps with C-O stretching band of ethoxy at 1018 cm^{-1} , so we assign the band at 1018 cm^{-1} to both ethoxy and*

phosphate.

In the phosphate buffer solution, vibrational band at 3046 cm⁻¹, 2920 cm⁻¹ and 2860 cm⁻¹ appeared at ca. 0 V_{RHE}, which is also observed at the potential values of 0.04 V_{RHE} and 0.07 V_{RHE} obtained in 0.1 M NaClO₄ solution. These bands were assigned to adsorbed vinyl alcohol and ethoxy intermediate, respectively. As a result, we can conclude that the same adsorbed intermediates are present in the phosphate buffer solution at a similar electrode potential on Cu.

4. Supplementary Figure 20 (in-situ Raman spectra between -0.4 to -0.9 V) should be merged with Figure 3.a (in-situ Raman spectra between 0.3 to -0.3 V) and presented in the main text. The information in Supplementary Figure 20 is indispensable to conclude that “vinyl alcohol and ethoxy intermediate present on Cu surface during the reduction of acetaldehyde”. In addition, the authors are suggested to acquire the in-situ Raman spectra beyond -1.2 V where the formation of ethane and ethylene starts. Due to the restricted mass transport of a reactant from solution, the surface vinyl alcohol signal is expected to vanish in the Raman spectra against negatively shifted potential. This expected observation will be decisive to conclude that the precursor of ethane and ethylene is the surface vinyl alcohol, and the surface vinyl alcohol is limited by the mass transport from the bulk solution. If the Raman band of surface vinyl alcohol still intensifies beyond -1.2 V, a specific explanation of the potential dependent band intensity should be given, otherwise, the kinetic model given by the reaction mechanism may be questionable.

We thank the reviewer for this comment, our Raman spectra has been merged as shown in Fig. R8, which is also added to the main text.

Fig.R8 | In-situ Raman spectra obtained on polycrystalline copper powder electrode as a function of potentials in Argon saturated 0.1 M NaClO₄ (pH=7) with 100 mM acetaldehyde.

We agree with the reviewer that surface vinyl alcohol signal is expected to vanish as electrode potential negative of -1.2 V is applied. However, we can not collect in situ Raman spectra beyond -0.9 V due to the accumulation of hydrogen bubbles near the electrode surface in the cell. During the potential window we investigated, as shown in **Fig.R2**, we observed that the band around 3045 cm⁻¹ start to decrease from -0.6 V_{RHE}, which was previously assigned to =C-H stretching of adsorbed vinyl alcohol, this result is in reasonable agreement with our RRDE measured onset potential (ca. -0.7 V_{RHE}) for reducing vinyl alcohol to ethylene and ethane on Cu powder electrode. Moreover, we observed an opposite trend for hydrocarbon related bands (-CH₃ and -CH₂) between 2800 cm⁻¹ and 3000 cm⁻¹. These bands tend to increase as the potential is stepped to more negative values of -0.6 V_{RHE}, possibly due to the accumulation of reduction products near the electrode surface at high reducing potentials. Thus, we can conclude that this vinyl alcohol related Raman signal starts to diminish when potential is beyond -0.6 V. This discussion is added to the main manuscript.

Fig.R2 | Additional in situ Raman spectra at high frequency region obtained on polycrystalline copper as a function of potentials in Argon saturated 0.1 M NaClO₄ (pH=7) with 100 mM acetaldehyde. All the potentials are on the RHE scale.

5. As for the Question 4 of Reviewer 3, the authors should note that there are four possible kinetic models that should be separately discussed. (1) For model 1, tautomerization in solution is much slower than the transport of acetaldehyde or enol to the surface, and acetaldehyde is the direct reaction precursor, it undergoes a catalytic step to be transformed into adsorbed enol on Cu. In this case, there will be a concentration gradient of acetaldehyde in the diffusion layer, whereas the enol concentration is constant in the diffusion layer. (2) For model 2, tautomerization in solution is even faster than the transport of acetaldehyde or the enol to the surface, and acetaldehyde is the reaction precursor, it undergoes a catalytic step to be transformed into adsorbed enol on Cu. In this case, there will be a concentration gradient of both enol and acetaldehyde in the diffusion layer. (3) For model 3, tautomerization in solution is much slower than the transport of acetaldehyde or the enol to the surface, and enol is the reaction precursor. In this case, there will be a concentration gradient of enol in the diffusion layer, whereas acetaldehyde concentration is constant in the diffusion layer. (4) For model 4, tautomerization in solution is even faster than the transport of acetaldehyde or the enol to the surface, and enol is the reaction precursor. In this case, there will be concentration gradients of both enol and acetaldehyde in the diffusion layer.

In model 1 or 3, only the direct precursor (either acetaldehyde or enol) contributes to the reaction current, whereas in model 2 or 4, due to the fast tautomerization in solution, both enol and acetaldehyde contribute to the reaction current. Bear these

in mind, we can infer that for each model, there will be a mass-transport limiting current when the surface concentration of the reaction precursor drop to 0 at RDE mode. The current value is expected to be proportional to the gradient of the corresponding reactant concentration (acetaldehyde concentration for model 1, enol concentration for model 3, and sum concentration for models 2 and 4), the authors are suggested to evaluate the mass-transport limiting current values for different models and compare them with the measured values to determine the true kinetic model.

We thank the reviewer for this careful and detailed suggestion. We agree with reviewer that a comprehensive numerical evaluation of mass-transport limiting current values for different models, and comparing them with experimental values, will provide a more rigorous analysis to determine the kinetic model in this study. However, there are several challenges in conducting the experiments as the reviewer mentioned. First, a completely sealed electrochemical cell compatible with RDE setup must be used to accurately quantify the concentration of acetaldehyde and vinyl alcohol due to small aldehydes are highly volatile under the ambient condition(<https://doi.org/10.1002/anie.201508851>). Second, a real-time deconvolution for HER, acetaldehyde and vinyl alcohol reduction currents must be achieved to exclude the interference from HER which is hard to avoid in aqueous media. While our current system does not allow us to fully seal the cell, our future work will focus on designing a completely sealed standard 3-electrode electrochemical cell compatible with RRDE, which can be used to acquire more reliable quantitative data to determine the kinetic model. This information has been added to the section of conclusions and future work in the main manuscript.

Reviewer #3 (Remarks to the Author):

The authors have well addressed all points I brought up in review, as well as those of the other reviewers. Revision has either clarified concerns (especially with respect to assignments of oxidation features on Pt to ethylene and ethane) or made clear limitations of the work so they are apparent to all readers. As such, I recommend publication of the revised work. I still believe this is a stronger contribution if the DFT portions of the study were removed, as the limitations of the approaches used lead to mechanistic conclusions being better supported by experimental data, with the DFT portions added more confusion due to these limitations than usefulness in supporting conclusions. However, these limitations are apparent to a knowledgeable reader, and the DFT work is not misrepresented with limitations discussed, if the authors still want to include this portion of their collaborative work.

We thank the reviewer's positive response and support of our work. To avoid bringing any confusion to readers from different communities, we have added following clarification to *the computational part in the main text*:

- 1. It is important to note that we ignored the potential dependent interaction of water with the Cu surface, therefore, our computational results may not provide an accurate estimation of adsorption energetics of vinyl alcohol at each electrode potential. Ab initio molecular dynamics (AIMD) simulations of cations in contact with the Cu surfaces with explicit water molecules would be needed to accurately model the possible potential dependent water displacement reaction, which is beyond the scope of this study. (page 14, included in last revision)*
- 2. It is important to note that pre-exponential factor and explicit solvent effect are not included in these computations, thus, these computational results must be compared with experimental results to draw any reliable conclusion. (page 18, included in current revision)*

REVIEWERS' COMMENTS

Reviewer #1 (Remarks to the Author):

In the revised manuscript, the authors have provided additional details and added further discussion. It not only makes the innovation of the current paper clearer but also demonstrates the limitations of the present study, which facilitates subsequent researchers to target their work. I, therefore, recommend this work for publication in Nature Communications. In addition, I would like the author to make a few minor improvements before the final publication.

1. In the response to Reviewer 2, the authors mention that Cu can catalyze the reduction of vinyl alcohol to ethoxy intermediate at -0.3 V vs. RHE. Could the authors add some citations to make the conclusion more reliable?
2. For the question 3 of Reviewer 2, the authors only explain the assignment of peaks without explaining the trend of peak intensity. In my opinion, this may be because in-situ Raman (even surface enhanced Raman) actually detects local species (rather than absolute surface-adsorbed species), and changes in local pH may induce the vibration of peak intensity (DOI: 10.1021/jacs.9b07000).

Reviewer #2 (Remarks to the Author):

The authors have addressed my concerns satisfactorily by carrying out additional experiments, presenting more data and reasonable discussions. Therefore, I recommend its acceptance as is.

Reviewer #1 (Remarks to the Author):

In the revised manuscript, the authors have provided additional details and added further discussion. It not only makes the innovation of the current paper clearer but also demonstrates the limitations of the present study, which facilitates subsequent researchers to target their work. I, therefore, recommend this work for publication in Nature Communications. In addition, I would like the author to make a few minor improvements before the final publication.

We thank the reviewer's positive response and support our work for publication.

1. In the response to Reviewer 2, the authors mention that Cu can catalyze the reduction of vinyl alcohol to ethoxy intermediate at -0.3 V vs. RHE. Could the authors add some citations to make the conclusion more reliable?

We have added several computational results on Cu(100) (<https://doi.org/10.1021/acscatal.5b01967>; <https://doi.org/10.1021/acscatal.1c01486>) to supplementary note 1 to support our conclusion, which is also copied below:

... This result is consistent with previous computational studies on Cu(100)^{9,10} which show that electroreduction of adsorbed vinyl alcohol to ethoxy is thermodynamically downhill at -0.3 V vs RHE...

2. For the question 3 of Reviewer 2, the authors only explain the assignment of peaks without explaining the trend of peak intensity. In my opinion, this may be because in-situ Raman (even surface enhanced Raman) actually detects local species (rather than absolute surface-adsorbed species), and changes in local pH may induce the vibration of peak intensity (DOI: 10.1021/jacs.9b07000).

Thank you for the suggestion, we have added this discussion into supplementary note 2, which is also copied below:

... It is important to note that local pH changes may result in the variation of intensities of phosphate bands because in situ Raman detects not only adsorbed species but also local species near the electrode surface¹²

Reviewer #2 (Remarks to the Author):

The authors have addressed my concerns satisfactorily by carrying out additional experiments, presenting more data and reasonable discussions. Therefore, I recommend its acceptance as is.

We thank the reviewer's positive response and support for our work.